# CAD-Tokenizer: Towards Text-based CAD Prototyping via Modality-Specific Tokenization

**Ruiyu Wang**[1]*, **Shizhao Sun**[2]†, **Weijian Ma**[3], **Jiang Bian**[2]
[1]University of Toronto, [2]Microsoft Research Asia, [3]Fudan University
rwang@cs.toronto.edu, shizsu@microsoft.com

## Abstract

Computer-Aided Design (CAD) is a foundational component of industrial prototyping, where models are defined not by raw coordinates but by construction sequences such as sketches and extrusions. This sequential structure enables both efficient prototype initialization and subsequent editing. Text-guided CAD prototyping, which unifies Text-to-CAD generation and CAD editing, has the potential to streamline the entire design pipeline. However, prior work has not explored this setting, largely because standard large language model (LLM) tokenizers decompose CAD sequences into natural-language word pieces, failing to capture primitive-level CAD semantics and hindering attention modules from modeling geometric structure. We conjecture that a multimodal tokenization strategy, aligned with CAD's primitive and structural nature, can provide more effective representations. To this end, we propose CAD-Tokenizer, a framework that represents CAD data with modality-specific tokens using a sequence-based VQ-VAE with primitive-level pooling and constrained decoding. This design produces compact, primitive-aware representations that align with CAD's structural nature. Applied to unified text-guided CAD prototyping, CAD-Tokenizer significantly improves instruction following and generation quality, achieving better quantitative and qualitative performance over both general-purpose LLMs and task-specific baselines.

## 1 Introduction

Computer-Aided Design (CAD) plays a central role in industrial prototyping and production. CAD models are constructed as sequences of operations—such as sketches and extrusions—that CAD kernels compile into machine-interpretable 3D objects. Unlike other 3D representations, CAD explicitly encodes design history, which makes it particularly suitable for both creating initial prototypes and performing subsequent modifications. Automating CAD sequence generation, especially from natural language instructions, could substantially accelerate the design workflow.

Recent work has begun to explore text-guided CAD subtasks, including CAD editing (Yuan et al., 2025) and text-to-CAD generation (Wang et al., 2025a; Li et al., 2025b; Khan et al., 2024b). These studies not only demonstrated the feasibility of each individual task but also showed that large language models (LLMs) can effectively operate on CAD sequences. However, real-world design is rarely a one-shot process: engineers iteratively refine prototypes by alternating between creation and modification. An effective text-based CAD system should therefore support both tasks in a unified manner—generating CAD objects from instructions while also allowing editing of existing designs.

Handling both tasks within a single unified model has never been studied previously. The difficulty lies in the distinct requirements of the two subtasks. Text-to-CAD demands strong *generative* capability to produce high-quality sequences from scratch, whereas CAD editing requires precise instruction *understanding* and alignment with existing geometry. A unified framework must therefore master two complementary but non-overlapping skills, which places significantly higher demands on the backbone than solving either task alone.

---

*Work done during the authors' internship at Microsoft Research Asia.
†Correspondence Author.

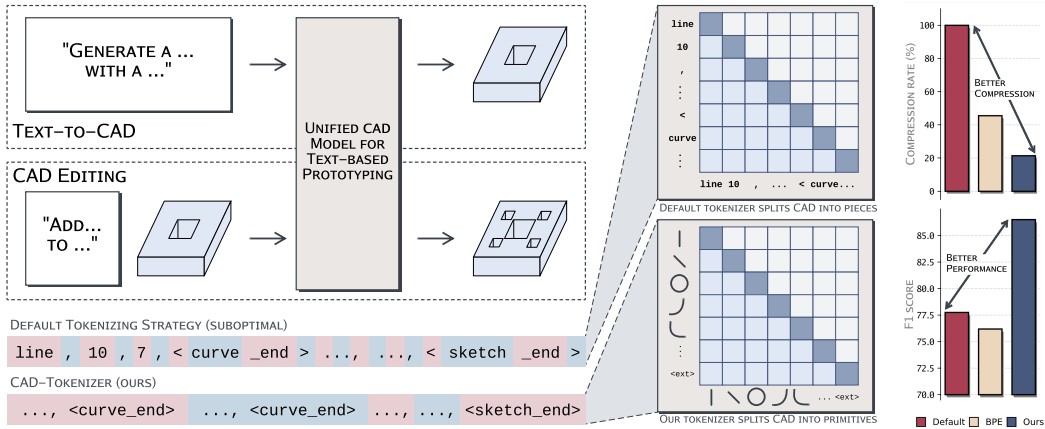

Figure 1: Overview of the **unified** text-based CAD prototyping task, which includes the Text-to-CAD generation and text-based CAD Editing. Our approach introduces a CAD-specific tokenization method that produces primitive-level tokens instead word pieces. This paradigm allows LLMs to capture relationships among CAD primitives more effectively, improves compression and performance.

We introduce CAD-Tokenizer, the first framework to tackle unified text-based CAD prototyping with CAD-specific tokenization. Our key insight is that the default tokenizers used in LLMs are poorly matched to CAD data. They decompose CAD programs into natural-language word pieces, fragmenting operations and numeric parameters into arbitrary substrings (e.g., "`[extru] [-sion]`"). This causes attention layers to focus on punctuation or partial tokens instead of meaningful primitives, limiting the model's ability to capture structural and geometric dependencies. To address this, we use a VQ-VAE to compress CAD sequences into primitive-level tokens before training, enabling LLMs to directly model inter-primitive relations and predict the next operation rather than the next character or word piece (Figure 1). This formulation makes unified training possible, leading to substantial improvements in both efficiency and quality.

Our contributions are as follows:

1. We develop a primitive-level VQ-VAE tokenizer that maps sketch–extrusion pairs into multiple discrete tokens. To integrate with LLMs, we train adapters that align the pretrained tokenizer vocabulary with the backbone's embeddings.

2. We fine-tune an LLM on these CAD-specific tokens under a unified setup covering both Text-to-CAD generation and CAD editing. This design improves training efficiency and surpasses strong task-specific baselines.

3. We leverage the formal grammar of CAD by introducing a finite-state automaton (FSA)–based sampling strategy, which enforces valid syntax in generation and improves quality.

4. Through extensive qualitative, quantitative, and ablation studies, we demonstrate that CAD-Tokenizer improves both efficiency and performance across unified CAD prototyping tasks.

## 2 RELATED WORKS

### 2.1 CAD MODELING

In general, CAD modeling tasks take different types of user inputs and generate sequential CAD output that renders into 3D objects by 3D kernels.

**Classical CAD Tasks.** Classical CAD tasks focus on generating CAD sequences from non-textual inputs. This includes hand sketches (Seff et al., 2022; Xu et al., 2025), point clouds (Khan et al., 2024a; Ma et al., 2024), sequence prefixes (Xu et al., 2022; 2023), or even random noise (Wu et al., 2021). There are also tasks involving masked variation, where some primitives are masked out and converted into random variants (Zhang et al., 2025; Xu et al., 2022; 2023). Our work differs

from these works in different ways. First, our work focuses on CAD generation from textual inputs. Second, while works in these categories often use transformer encoders to extract features and decode specific IDs, they do not unify the encoding procedure into one model due to technical limitations. Third, while some of the previous works (Xu et al., 2022; 2023) train transformer modules using their codebooks, they do not involve texts or natural language inputs and thus have no need to leverage pretrained LLMs. This prevents the need to add alignment modules in the classic paradigms.

**Text-based CAD Tasks.** Text-based CAD tasks focus on generating CAD sequences based on textual instructions. Inputs for these tasks fall into two main types: the first type includes only textual instructions on how to generate an initial CAD model, and the second type includes a CAD sequence (of the same representation format) and a textual instruction on how to modify the given model. The former is the Text-to-CAD task (Wang et al., 2025a; Li et al., 2024; 2025b; Khan et al., 2024b; Govindarajan et al., 2025; Guan et al., 2025; Wang et al., 2025b) and the latter is the CAD-editing task (Yuan et al., 2025). Our work differs from these tasks as we propose the text-based CAD prototyping task, which merges them. Given that initiating and editing CAD models is a standard procedure in industry, a single-model solution is both necessary and promising for the prototyping pipeline.

## 2.2 LLM Tokenization

LLMs operate over discrete tokens that are mapped into hidden embeddings. In natural language, these tokens are defined by vocabularies (Pennington et al., 2014; Mikolov et al., 2013) or obtained via algorithms such as byte-pair encoding (BPE) (Gage, 1994). In multimodal tasks where inputs often lie outside the space of predefined vocabularies, external encoders and decoders such as VAEs and VQ-VAEs (Kingma & Welling, 2022; van den Oord et al., 2018), vision transformers (Dosovitskiy et al., 2021; Radford et al., 2021), or diffusion-based algorithms (Podell et al., 2023) are commonly used to map continuous modalities into token sequences that LLMs can process.

Tokenization strategies vary across domains. For visual tasks, pretrained VAE/VQ-VAE or CLIP/ViT encoders are typically used, sometimes paired with custom decoders when generative outputs are required (Zhou et al., 2024; Ge et al., 2023; 2025; Liu et al., 2023). In robotics, many works bypass neural tokenizers, instead stringifying actions for direct use with default LLM tokenizers (Brohan et al., 2023b;a), or applying lightweight algorithmic encodings (Pertsch et al., 2025). Time-series modeling has explored both neural tokenizers based on attention architectures (Nie et al., 2023) and classic tokenization methods (Li et al., 2025a).

Prior CAD works are task-specific on tokenizations. Classical CAD generation methods train separate encoders and decoders for different inputs, such as sketches and extrusions (Xu et al., 2022; 2023; Khan et al., 2024b;a), resulting in multi-encoder pipelines. Recent text-based CAD efforts typically rely on off-the-shelf LLM tokenizers, directly applying LLM tokenizations to CAD sequences without adaptation (Wang et al., 2025a; Yuan et al., 2025; Zhang et al., 2025). In contrast, our work introduces the first encoder–decoder tokenizer specifically designed to integrate with LLMs. We employ a single VQ-VAE–variant encoder–decoder pair to tokenize all CAD inputs and align its vocabulary with the LLM backbone. This unified tokenization allows the model to reason over CAD primitives as discrete units, addressing the fragmentation issues inherent in natural-language tokenizers.

## 3 Methodology

### 3.1 Approach Overview

The text-based CAD prototyping task unifies Text-to-CAD generation and CAD editing. Formally, given a prompt $x = (\mathcal{I}, C_{\text{orig}})$ consisting of an instruction $\mathcal{I}$ and an optional original sequence $C_{\text{orig}}$, the goal is to learn a mapping $f(\cdot)$ that outputs a parametric sequence $C_{\text{gen}}$. For Text-to-CAD, $C_{\text{orig}} = \emptyset$, while for editing tasks, $C_{\text{orig}}$ is a valid CAD sequence. The generated sequence should reflect the design intent expressed by the instruction and, when applicable, the original shape.

CAD-Tokenizer addresses this unified task by fine-tuning LLMs with CAD-specific tokens produced by a pretrained VQ-VAE (Figure 2). Specifically, (1) a primitive-based VQ-VAE is trained on CAD sequences to compresses the sketch-extrusion pairs into discrete tokens, and adapters are deployed to align these tokens with the backbone's embedding space (Section 3.2), (2) the backbone is fine-tuned

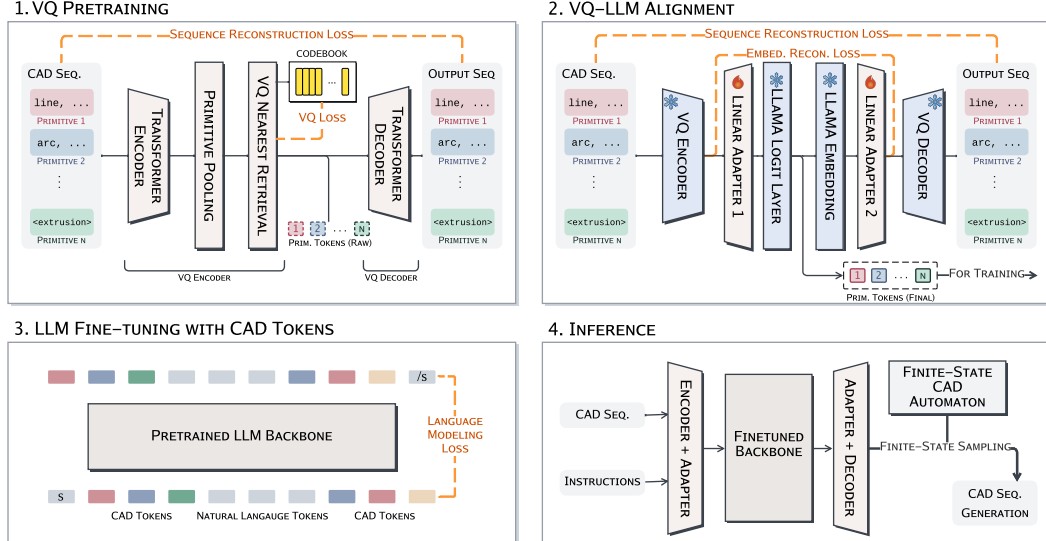

Figure 2: Overview of the CAD-Tokenizer framework. (1) A primitive-based VQ-VAE tokenizes CAD sequences into discrete primitive tokens. (2) Adapter modules align these tokens with the LLM embedding space. (3) The backbone LLM is fine-tuned using CAD-specific tokens. (4) At inference, an FSA-guided sampling strategy ensures syntactically valid CAD generation.

on the encoded CAD data (Section 3.3), and (3) an FSA-based decoding strategy enforces CAD grammar during inference (Section 3.4). We describe each step below.

## 3.2 PRIMITIVE-BASED VQ-VAE AS CAD TOKENIZER

**Pretraining Primitive VQ-VAE.** To convert CAD sequences into compact symbolic representations, we modify and pretrain a VQ-VAE variant that outputs primitive-level tokens. Each input sequence $C$ needs to be decomposed into sketch–extrusion pairs $SE = \{t_1, \dots, t_n\}$ and encoded into latent vectors $\{p'_1, \dots, p'_k\}$, with $k < n$, in pairs by transformer encoders. Unlike standard VQ-VAEs that pool the entire sequence into a single latent vector, we introduce a primitive-specific pooling layer, producing multiple discrete representations for each sketch–extrusion pair. This design captures both local and contextual information from the surrounding sequence. Training follows a sequence reconstruction objective with an accumulated VQ loss across all pooled tokens:

$$\mathcal{L}_{VQ\text{-}Prim} = \sum_{i \in n} \text{EMD}(Decoder(\{p'\}, t_{1,i-1}), \mathbb{1}_i) + \sum_{j \in k} VQ(\overline{T}_j^E, p'_j),$$

where EMD is the squared Earth Mover's Distance Loss, $\{p'\}$ denotes the set of all primitive latent vectors $\{p'_1, ..., p'_k\}$ generated by the encoder and pooling layer, $\mathbb{1}_i$ denotes the input data's one-hot property, and $\overline{T}_j^E$ denotes the pooled vector of the tokens construct up the $j$-th primitive. We demonstrate this method in Figure 2(1) and provide additional details such as the detailed pooling procedure in Appendix B.

**Aligning CAD Tokenizer with LLM Embeddings.** The raw primitive vectors $\{p'_j\}$ have dimension $d_{vq}$, while the LLM requires tokens in its embedding space of dimension $d_{tok}$. Typically, this adaptation is performed during co-training with LLM backbones (Liu et al., 2023), which is compute-intensive and does not guarantee alignment in the reverse direction. We introduce a novel use of adapters that map VQ-VAE outputs and LLM embeddings bi-directionally. In detail, we leverage the frozen *embedding* and *logit* layers of the pretrained LLM. Adapters are trained with a vector reconstruction loss that encourages the mapped tokens $\{p_j = \text{argmax}(L_{logit}(W_{d_{tok}}^{d_{vq}}(p'_j)))\}$ to remain faithful to their raw representations:

$$\mathcal{L}_{\text{recon}} = \sum_{j \in k} \|\hat{p}'_j - p'_j\|_2^2 + \|L_{embed}(p_j) - W_{d_{tok}}^{d_{vq}}(p'_j)\|_2^2.$$

$\hat{p}'_j = W^{d_{tok}}_{d_{vq}}(L_{embed}(p_j))$ is the reconstructed embedding corresponding to $\{p'_j\}$. The latter term is the reconstruction of the post-embedding layer tokens and the pre-logit tokens to ensure both adapter layers are trained. This alignment produces native LLM-recognizable primitive IDs, enabling seamless integration without modifying the backbone. The procedure is illustrated by Figure 2(2).

### 3.3 INSTRUCTION TUNING WITH CAD TOKENS

With aligned tokens, CAD sequences can be encoded into compact symbolic forms and directly used for instruction tuning. Given an input prompt $x = (\mathcal{I}, C_{\text{orig}})$ and target sequence $C_{\text{gen}}$, we define $x' = (\mathcal{I}, \text{CAD-Encoder}(C_{\text{orig}}))$ and $y' = \text{CAD-Encoder}(C_{\text{gen}})$. We fine-tune the pretrained LLM by minimizing the standard cross-entropy loss between the generated tokens $\hat{y} = f(x')$ and ground-truth $y'$ as described below:

$$\mathcal{L}_{SFT} = -\mathbb{E}_{(x',y')}\left[\frac{1}{T}\sum_{t=1}^{T}\log p(\hat{y} = y'_t|x')\right].$$

Because CAD sequences are significantly compressed, i.e., $|x'| < |x|$, fine-tuning is not only more effective but also more computationally efficient. Figure 2(3) illustrates this stage.

### 3.4 MODEL INFERENCE AND TOKEN DECODING

During inference, naive autoregressive sampling may produce invalid CAD sequences, as standard LLM decoding strategies (e.g., top-$p$, beam search) are unaware of CAD grammar. However, unlike natural language, CAD sequences are formal languages and can be fully described by states and transitions. Therefore, we can improve the sample quality by designing a finite-state automaton (FSA) that formalizes valid CAD construction rules (Figure 2(4)).

At each step, the FSA provides a series of masks restricting the candidate logits to grammar-compliant tokens, ensuring syntactic validity. The automaton state evolves with the decoder, whose choice for the next token is also passed to the automaton for deciding if a transition is going to take place. Despite some context-dependent geometric constraints are unable to fully catched, this method significantly reduces syntactic errors and improves sampling robustness by preventing obvious grammatical mistakes. We demonstrate the detailed process by Algorithm 1 and the FSA design in Figure 3. Further implementation details are given in Appendix B.2.

---

**Algorithm 1** The FSA-driven decoding algorithm.

---

Given: LLM-generated primitive tokens $T$
Given: Max length of generation $max\_len$
$State \leftarrow$ `init`
$M \leftarrow [\texttt{initial\_mask}]$
$Seq \leftarrow \emptyset$
**while** $|Seq| < max\_len$ **do**
  $m \leftarrow M.\text{next}()$
  logits $\leftarrow$ Decoder($Seq, T$)
  choice $\leftarrow$ argmax($m \otimes$ logits)
  $Seq.\text{append}(\text{choice})$
  **if** $M = \emptyset$ **then**
    new\_state, new\_M = FSA($State$, choice)
    $State \leftarrow$ new\_state
    $M \leftarrow$ new\_M
  **end if**
**end while**

---

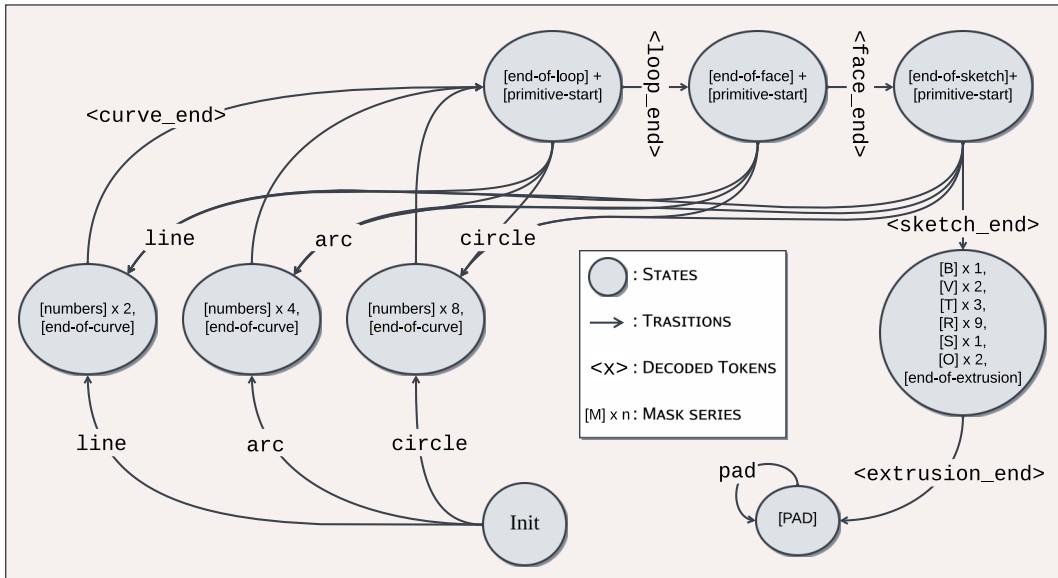

Figure 3: An overview of our FSA design. At each step, the FSA receives an input action, transitions to the corresponding new state (i.e., updates its internal state), and returns the mask(s) associated with that new state (node). The FSA map is specific to our CAD sequence representation, which can be found in Appendix A.

## 4 EXPERIMENTS

### 4.1 EXPERIMENTAL SETUPS

**Datasets.** For training the VQ-VAE, we convert the SkexGen (Xu et al., 2022) dataset into the formatting used by CADFusion (Wang et al., 2025a) and CAD-Editor (Yuan et al., 2025). We include the formatting details in the Appendix A. To prevent a distribution gap for the tokenizer between seen and unseen data, we use half of the training data for VQ-VAE training, ensuring the LLM also encounters samples not seen by the VQ-VAE encoder during its training. For training the backbone LLM, we use the datasets of CADFusion and CAD-Editor. Although both originate from SkexGen, the two datasets have different content, so we merge them for training our model without further deduplication. Since CAD-Editor uses the original split of SkexGen but the entire CADFusion dataset is split from its training set, we consolidate the CADFusion dataset and additionally annotated 200 pairs for our test set. We also balance this combined dataset by randomly selecting five CAD-Editing data points for every Text-to-CAD data point. Ultimately, this results in approximately 100k training pairs and 1000 testing pairs.

**Implementation Details.** The CAD sequence formalization and primitive selection is detailed in Appendix A. The VQ-VAE is trained with 5 encoder and 5 decoder layers, a codebook size of 2048, and an internal dimension of 512. It is trained with a learning rate of 3e-4 for up to 250 epochs, using early stopping with a patience of 25. The logit and embedding layers inserted are from `LLaMA-3-8b` (4096-dim), which is also used as our LLM backbone. The alignment uses exactly the same setup, but the encoder, decoder and LLM layers are frozen, and only the two adapter layers are trained. For LLM training, the backbone is configured with a maximum token length of 1024 and a hidden dimension of 4096. We adopt Low-Rank Adaptation (LoRA) (Hu et al., 2021) for efficient fine-tuning, with a rank of 32 and an $\alpha$ value of 32. The model is trained for 20 epochs with a learning rate of 1e-4 and the AdamW optimizer. The VQ-VAE training is run on a single NVIDIA A100-80G GPU, and the LLM fine-tuning is run by four NVIDIA A6000-48GB SMX GPUs using Distributed Data Parallel (DDP).

**Baselines.** We select two types of baselines. First, we include **CADFusion** and **CAD-Editor** as state-of-the-art methods in Text-to-CAD and CAD-editing tasks, respectively. However, since these are for individual text-based tasks only and no existing works address text-based CAD prototyping, we propose two additional baselines for the unified task. First, we train a vanilla LLM model using

| Methods | F1-Skt | F1-Ext | CD | COV | JSD | MMD | IR | VLM | HE |
|---|---|---|---|---|---|---|---|---|---|
| **CAD-Editor**[*] | 73.3 | 82.6 | 40.7 | 51.1 | **1.63** | **1.25** | **1.50** | 4.28 | **1.63** |
| **GPT-4o** | 80.0 | 78.1 | 42.9 | 51.8 | 4.00 | 2.14 | 47.9 | 1.94 | - |
| **Vanilla-LLaMA** | 78.8 | 84.9 | 42.4 | 48.6 | 3.90 | 2.37 | 48.6 | 4.31 | 2.64 |
| **CAD-Tokenizer** | **88.6** | **94.8** | **13.5** | **52.4** | 2.54 | 1.97 | 8.38 | **5.09** | 1.72 |

(a) **CAD-Editing** quantitative results. [*]**CAD-Editor** is the task-specific model on it.

| Methods | F1-Skt | F1-Ext | CD | COV | JSD | MMD | IR | VLM | HE |
|---|---|---|---|---|---|---|---|---|---|
| **CADFusion**[†] | 68.8 | 80.1 | 38.5 | 54.4 | 10.6 | 3.86 | 22.5 | **5.41** | 1.91 |
| **GPT-4o** | 66.7 | 66.8 | 79.8 | 52.6 | 62.0 | 13.5 | 90.5 | 1.47 | - |
| **Vanilla-LLaMA** | 66.4 | 80.9 | 48.4 | 53.8 | 27.4 | 72.6 | 80.5 | 3.45 | 2.58 |
| **CAD-Tokenizer** | **77.9** | **84.7** | **26.7** | **54.5** | **7.26** | **3.74** | **1.50** | 3.82 | **1.62** |

(b) **Text-to-CAD** quantitative results. [†]**CADFusion** is the task-specific model on it.

Table 1: Results in (a) and (b) include F1 scores for sketches (**F1**-skt ↑) and extrusions (**F1**-ext ↑), Chamfer Distance (**CD** ↓), Coverage(**COV** ↑), Minimum Matching Distance (**MMD** ↓), Jensen-Shannon Divergence (**JSD** ↓), Invalidity Ratio (**IR** ↓), the VLM Score (**VLM** ↑), and average rank from Human Evaluation (**HE** ↓). GPT-4o generations were not included in the human evaluation process. (↑) marks the higher the better, and (↓) marks the opposite. The best results are marked in **bold**, and second-best results are underlined.

the native LLaMA tokenizer (**Vanilla-LLaMA**), with all other aspects of the setup identical to CAD-Tokenizer. Second, we prompt a five-shot **GPT-4o** to examine how modern commercial LLMs perform on the text-based CAD prototyping task. Details including baseline implementation and prompting samples can be found in the Appendix C.1.

**Metrics.** Our evaluation focuses on different aspects of the generated CAD models. First, we measure the pairwise correspondence between generated outputs and the ground truth sequences. For this criterion, we use **F1** scores and Chamfer Distance (**CD**). We follow CADFusion's (Wang et al., 2025a) measurements of **F1-Sketch** and **F1-Extrusion** for simplicity. Second, we measure distributional similarity. Between the generated samples and the ground truths, we follow (Xu et al., 2022; Yuan et al., 2025; Wang et al., 2025a) and measure: Coverage (**COV**), which quantifies how well the generated distribution covers the ground truth; Minimum Matching Distance (**MMD**), which evaluates similarity by finding the closest matches between samples from the two distributions; and Jensen-Shannon Divergence (**JSD**) for distributional similarity. Third, we measure the Invalidity Ratio (**IR**) as an indicator of sequence robustness; and fourth, we measure the generated outputs' alignment with instructions using a VLM-based score (**VLM**) and Human Evaluation (**HE**). The VLM is prompted to rate instruction following on a scale of 0 to 10, and human judges are asked to rank the outputs from different models based on the same criterion. Details on VLM prompting and human evaluation methodology can be found in Appendix C.1.

For the clarity of the presentation, the results of the distribution scores (**COV, JSD, MMD**) and chamfer distance in all records are multiplied by $10^2$.

## 4.2 MAIN RESULTS

**Quantitative Evaluation.** Table 1 reports results on CAD editing (a) and Text-to-CAD generation (b). On CAD editing, CAD-Tokenizer achieves the best results across almost all metrics. It improves the F1 scores by around 10 points each. Compared to the task-specific CAD-Editor, CAD-Tokenizer leads in F1 metrics while also achieving stronger VLM and human evaluation scores. Moreover, discoveries detailed in the supplementary results of Appendix C.2.1 suggest that our model is best intended to edit the original object despite being disfavored by the distributional metrics by doing so.

On Text-to-CAD generation, CAD-Tokenizer again shows clear improvements. It significantly improves the F1 scores, indicating strong matching in CAD sequential information. Chamfer Distance is also decreased by a large margin. Although CADFusion attains a slightly higher VLM score (5.41 vs. 3.82), CAD-Tokenizer surpasses it in nearly every other metric, including a lower human evaluation rank that indicates preference.

| Methods | F1-Skt (↑) | COV (↑) | JSD (↓) |
|---|---|---|---|
| **HNC-CAD** | 85.5 | 57.5 | 29.8 |
| **CAD-Tokenizer** *(curve)* | **94.1** | **64.5** | **8.19** |
| **CAD-Tokenizer** *loop* | 91.5 | 59.5 | 18.4 |
| **CAD-Tokenizer** *single* | 76.5 | 54.0 | 35.9 |

Table 2: The ablation study on the reconstruction quality of the CAD-Tokenizer variants and **HNC-CAD**. Only the sketch scores are reported for the **F1** score because the CAD-Tokenizer encodes objects by sketch-extrusion pairs and always achieves full score for this sub-metric. **CAD-Tokenizer** *(curve)* is the default variant that we reported in main results.

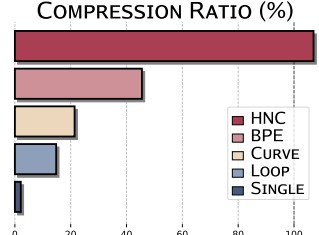

Figure 4: The Compression ratio of the different tokenization algorithms. The compression ratio is 100% for no compression.

Besides the advantage of our framework over task-specific baselines, it is noteworthy that Vanilla-LLaMA almost collapses on the majority of metrics. This supports our conjecture that text-based CAD prototyping is inherently a challenging task for conventional tokenizers, and highlights that our improvements over the vanilla tokenizer and training paradigm are substantial.

**Qualitative Evaluation.** Figure5 illustrates typical generations of our model and selected baselines. In Text-to-CAD, CAD-Tokenizer produces more faithful multi-face objects compared to Vanilla-LLaMA, which often drops faces or misaligns structures. Also, the outputs are natural and balanced, which resemble human designs. Its qualitative performance is competitive with CADFusion, the task-specific baseline.

For CAD editing, CAD-Tokenizer demonstrates stronger instruction-following: it modifies existing shapes as required rather than persisting the original form, a failure mode described in Appendix C.2.3. Compared to the vanilla tokenizer, CAD-Tokenizer better respects structural templates and achieves higher-quality modifications.

Additional results including more complex prompts and designs can be found in Appendix C.2.2.

### 4.3 Ablation Studies

We analyze the contribution of each component in CAD-Tokenizer across three stages: pre-SFT tokenization, in-SFT integration with the LLM, and post-SFT sampling.

**Before LLM Finetuning: the Quality of Tokenizers.** The tokenizer quality is critical to downstream LLM performance. We evaluate two aspects: (i) **reconstruction quality**, in terms of selected sequence and distribution metrics, and (ii) **compression ratio** for efficiency. We compare CAD-Tokenizer against the following: 1. **HNC-CAD** (Xu et al., 2023), a strong neural coding baseline on conditional and unconditional CAD generation, 2.**BPE** (Gage, 1994), the Byte-pair Encoding, a standard tokenization algorithm, 3. **CAD-Tokenizer** *(curve)*, our default tokenizer with curve-based pooling, 4. **CAD-Tokenizer** *(loop)*, a variant pooling loops instead, and 5. **CAD-Tokenizer** *(single)*, which omits primitive pooling and encodes each sketch–extrusion pair as a single unit.

Table2 and Figure 4 presents the results. Except that BPE is not involved in the reconstruction evaluation as it guarantees returning original vectors algorithmically, the exploration is three-fold: First, all CAD-Tokenizer variants outperform prior methods on both reconstruction quality and compression. The CAD-Tokenizer variants perform uniformly better than other tokenization methods, while having better compression rate. Second, within CAD-Tokenizer, the results also highlight the trade-off between compression rate and the reconstruction accuracy, as the variants that compress more tokens into one embedding tend to have worse quality after reconstruction. Third, the sharp degradation of the *single* variant underscores the importance of primitive-level pooling.

**In LLM Finetuning: the Performance of Different Tokenizion Methods.** We next evaluate the tokenizers during supervised fine-tuning (SFT) with the LLM backbone. All methods are included except HNC-CAD, which was not designed for LLM training.

As shown in Table 3, both BPE and the *single* variant of CAD-Tokenizer perform poorly, with high invalidity ratios and consistently weak metrics. Their setups justify the unsatisfactory performance:

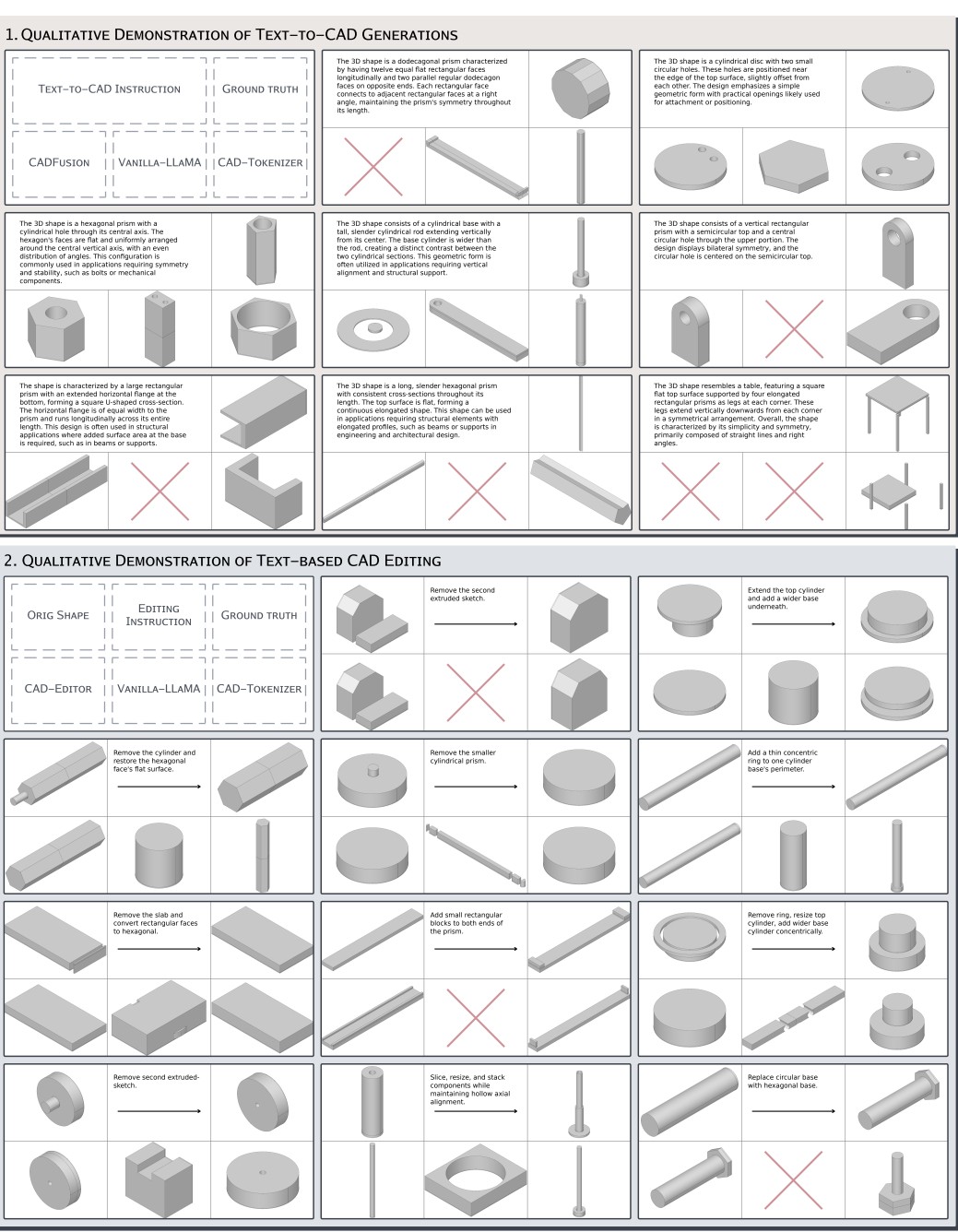

Figure 5: Qualitative results. The upper and lower sections display results for the Text-to-CAD and CAD editing tasks, respectively. In each subfigure, the top-left corner shows the input instruction and CAD object (or only the instruction in Text-to-CAD), the top-right corner shows the ground truth, and the bottom three show outputs from CADFusion/CAD-Editor, Vanilla-LLaMA, and CAD-Tokenizer.

the *single* variant performed already poor enough during the pre-finetuning reconstruction evaluation; and BPE does not align the post-tokenization tokens with the LLM vocabulary, which adds up additional burden during finetuning. The *loop* variant achieves performance close to the default *curve* tokenizer, with slightly lower reconstruction but higher compression. This not only indicates that primitive pooling—whether curve- or loop-based—is essential, but also offers flexibility and alternatives in trading off accuracy against efficiency.

| Methods | **F1**-Avg (↑) | **CD** (↓) | **COV** (↑) | **JSD** (↓) | **IR** (↓) |
|---|---|---|---|---|---|
| **CAD-Tokenizer** *(curve)* | **86.5** | **20.1** | 53.4 | **4.90** | 4.94 |
| **CAD-Tokenizer** *loop* | 86.3 | 32.6 | **53.6** | 7.05 | **4.91** |
| **CAD-Tokenizer** *single* | 78.3 | 59.1 | 43.4 | 48.5 | 70.7 |
| **BPE** | 76.2 | 56.8 | 48.8 | 48.3 | 88.5 |

Table 3: The ablation study of each tokenizer when integrated and trained with the backbone LLM. **CAD-Tokenizer** *(curve)* is the default variant which we reported in the main quantitative results, while *loop* is a variant on pooling loop primitives and *single* does not perform specific pooling.

| Methods | **F1**-Avg (↑) | CD (↓) | COV (↑) | JSD (↓) | IR (↓) |
|---|---|---|---|---|---|
| **CAD-Tokenizer**(*+FSA*) | **86.5** | **20.1** | **53.4** | **4.90** | **4.94** |
| **CAD-Tokenizer**+*top-p* | 80.4 | 66.4 | 30.3 | 61.2 | 17.2 |
| **CAD-Tokenizer**+*beam search* | 82.8 | 49.8 | 51.2 | 46.7 | 45.2 |

Table 4: The ablation study on the generation quality of each sampling method after training. The base model used for evaluation is the fine-tuned CAD-Tokenizer. **CAD-Tokenizer(*+FSA*)** is the default variant which we reported in the main quantitative results.

**After LLM Finetuning: Different Sampling Strategies.** Finally, we assess the impact of our FSA-based decoding strategy described in Section 3.4. We fix the backbone to the fine-tuned CAD-Tokenizer and apply different sampling strategies on top of it, and compare our method with the two mainstream approaches: top-$p$ sampling and *beam search*.

Table 4 shows that FSA-guided sampling consistently outperforms both alternatives across all metrics. Its CAD-specific design leverages formal sequence constraints, enabling it to utilize structural formatting information that general-purpose decoding methods do not take into account. Also notably, *beam* search improves the model performance in all aspects comparing to the top-$p$ sampling, but encounters a significant tradeoff in invalidity ratio.

## 5 LIMITATIONS

Our work's limitations are two-fold. First, the quality gap between the open-source and private-sector CAD data limits us from training on further complex shapes. Second, as mentioned and referred, we observe a gap between the distributional metrics and the actual performance in CAD Editing, which stems from the failure to evaluate the 'intent to keep the original shape.' We therefore anticipate both better datasets and metrics that can create a better ground for future works to improve. Additionally, our failure study in Appendix C.2.3 suggests better spatial or commonsense reasoning capabilities, which can only be improved by better pretrained backbones. We include some of the related discussions, as well as the detail of our failure study, in Appendix C.2.

## 6 CONCLUSION

In this work, we presented CAD-Tokenizer, the first framework to address the unified text-based CAD prototyping problem, a task that requires a unified model on both Text-to-CAD generation and CAD editing. Key to CAD-Tokenizer's approach is a pretrained transformer-based VQ-VAE module that enables CAD-specific tokenization by converting sequential CAD inputs into meaningful primitive tokens, instead of the word pieces produced by vanilla tokenizers. LLM backbones that were finetuned on the CAD specific tokens demonstrated better performance and training efficiency. Our extensive experiments validated these core design choices, including the primitive-specific pooling mechanism within the VQ-VAE and the FSA-driven sampling, and demonstrated the advantage of CAD-Tokenizer over both task-specific and general-purpose baselines. We also improved inference-time decoding quality by introducing a novel finite-state automaton–driven sampling method, designed to enforce sequence formatting and hence improve generation quality.

## ETHICS STATEMENT

The data used in this work is tailored for CAD generation. Due to its specialized nature, the misuse risk is naturally minimized, ensuring the developed methods primarily benefit CAD development.

## REPRODUCIBILITY STATEMENT

We have included sufficient information on our all aspects of our experimentations including dataset resources and setups, model hyperparameters and LLM backbones, computational resources and setups, and metric setups. Relevant detail can be found in Section 4.1 and Appendix C.1.

## THE USE OF LARGE LANGUAGE MODELS

The use of LLMs is restricted in revising the word choices and grammar only. No LLMs are used for research ideation or contributive enough to be regarded as a contributor.

## ACKNOWLEDGEMENTS

We would like to thank Lexin Zhou, Rushuai Yang, Yun Pan for insightful discussions. We appreciate the discussion with Zhuoyu Jiang and Yuheng Lu during the rebuttal phase. We also acknowledge the support from the MSR Asia Industry Innovation Center for motivating real-world examples and guidance on industrial pipelines.

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

## A  ADDITIONAL INFORMATION OF THE CAD REPRESENTATION

### A.1  SEQUENCE REPRESENTATION

We follow the practice of Zhang et al. (2025); Yuan et al. (2025); Wang et al. (2025a). In detail, a CAD object is defined by a series of sketches and extrusions, termed the *Sketch-and-Extrude Modeling (SEM)* format.

Each sketch consists of multiple faces, with each face typically containing one or more loops. Primitives in each loop include lines, arcs, and circles. They are defined by an identifier and, for their geometric properties, one, two, or four key coordinates or sets of parameters, respectively.

Each extrusion is represented as a `BVVTTTRRRRRRRRRSOO` sequence. `B` refers to the boolean operation selected from `add, cut, intersect`; the two `V` values define the respective displacements of the top and bottom extrusion planes from a reference plane; the three `T` values form the translation vector; the nine `R` values represent rotation parameters; `S` is the scaling factor; and the two `O` values identify the origin of scaling.

Figure 6 illustrates the components of the sketch and extrusion operations.

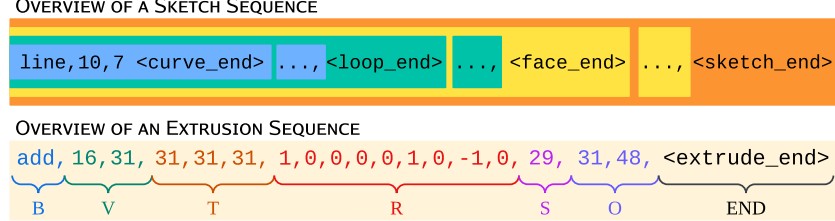

Figure 6: An overview of the CAD sequence structure. Top: sketch components; bottom: extrusion components.

In our VQ-VAE training, we tokenize CAD sequences into their smallest semantic units. These units are defined in Table 5. This tokenizer differs from native LLM tokenizers in that it does not further split these semantic units into word pieces.

| Token Name | Vocabulary Index | Description |
|---|---|---|
| *pad* | 0 | Padding token |
| *line* | 1 | Identifier token of curve: line |
| *arc* | 2 | Identifier token of curve: arc |
| *circle* | 3 | Identifier token of curve: circle |
| *<curve_end>* | 4 | End token of curves |
| *<loop_end>* | 5 | End token of loops |
| *<face_end>* | 6 | End token of faces |
| *<sketch_end>* | 7 | End token of sketches |
| *add* | 8 | Identifier token of extrusion operation: add |
| *cut* | 9 | Identifier token of extrusion operation: cut |
| *intersect* | 10 | Identifier token of extrusion operation: intersect |
| *<extrusion_end>* | 11 | End token of extrusions |
| *-1* | 12 | Number -1 |
| *0-63* | 13-76 | Numbers from 0 to 63 |

Table 5: Token vocabulary for the CAD sequence representation used in training our VQ-VAE.

### A.2  PRIMITIVE DEFINITION

We define our primitives differently for sketches and extrusions. For sketches, we define one primitive as a curve and any succeeding end tokens, if present. For extrusions, we split each into three parts:

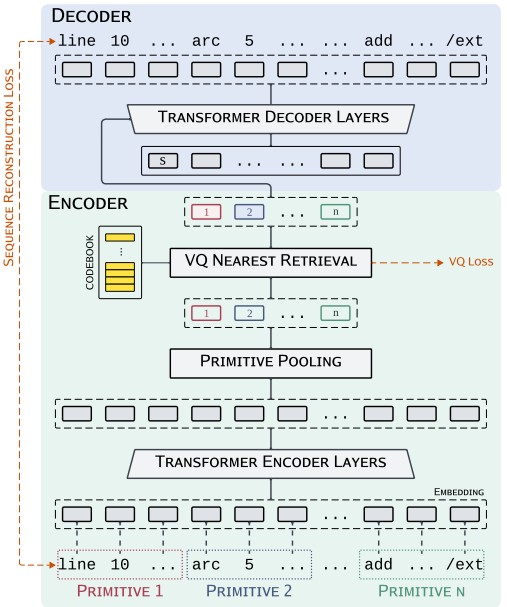
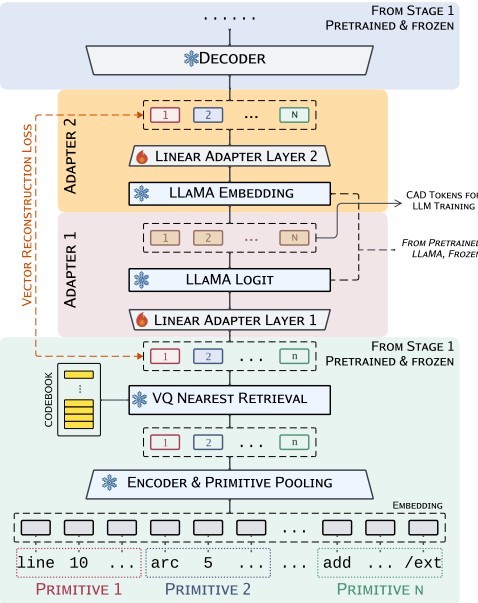

Figure 7: A detailed overview of the VQ-VAE pretraining and finetuning.

`B,V,T,R`, and `S,O,END`, and split `R` into 3 distinct tokens. This partitioning is designed to prevent the VQ-VAE from having to summarize overly long sequences, which can be difficult to learn effectively.

Our primitive design offers several advantages. First, limiting the scope to the loop level restricts the number of tokens the VQ-VAE model generates for the LLM to a manageable quantity. Generating one token per curve would provide the LLM with too many tokens; conversely, one token per face or per sketch would yield too few for the LLM to extract meaningful correlations and, in the case of per-sketch tokenization, might also represent too much information for a VQ-VAE to compress effectively into a single token. Second, compared to works that do not pool primitives but instead train on smaller 'particles' (Xu et al., 2022; 2023), our approach improves representation quality by embedding all primitive representations into a unified subspace (which is not guaranteed by methods training separate models for different CAD pieces) and by posing contextualization. Furthermore, those approaches may not adequately address the challenge of long extrusion sequences, which are difficult for a VQ-VAE to compress meaningfully into a single token.

# B  IMPLEMENTATION DETAILS

## PRELIMINARY: TERMINOLOGIES AND NOTATIONS

We use the term `Encoder` and `Decoder` to refer the *transformer encoder* and *decoder* layers in with in the VQ module. We refer the term `VQ Encoder` to the entire module before token extraction, i.e. the *transformer encoder layers*, the *primitive pooling*, and the *VQ layer*. The same applies to `VQ Decoder`, but since no additional modules are added, it is in fact equivalent to the `Decoder`. `CAD Encoder` and `CAD Decoder` describe the combination of the `VQ Encoder/Decoder` and the corresponding *adapter layers*.

## B.1  VQ AND PRIMITIVE POOLING

A classic VQ-VAE for sequential data consists of transformer encoder layers, a VQ layer to quantize vectors, and transformer decoder layers. Input tokens are encoded, pooled into a single hidden

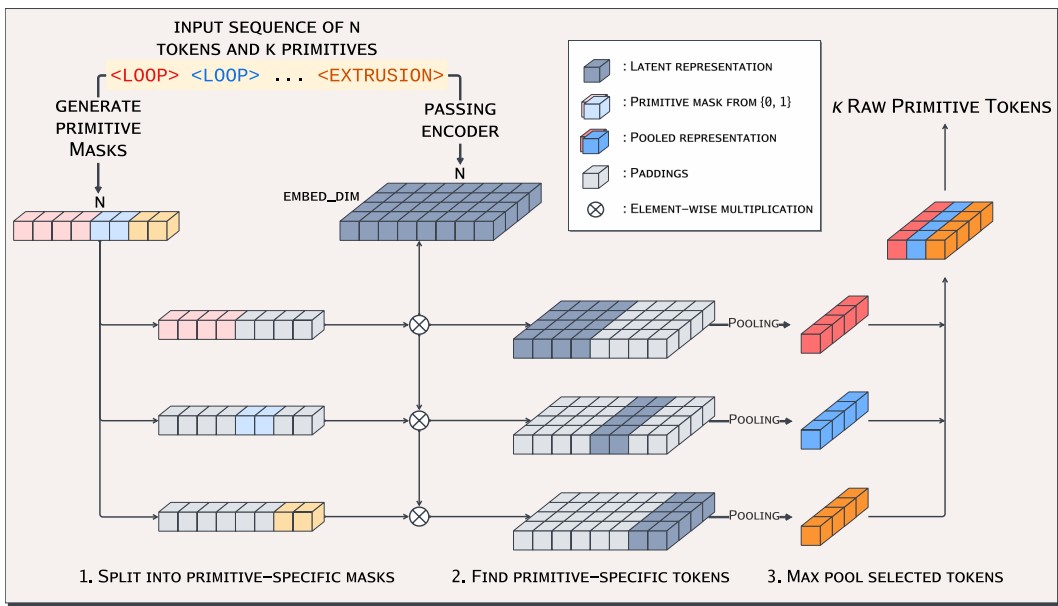

Figure 8: An overview of our primitive pooling procedure for each sequence.

vector, and then mapped to codebook vectors. These VQ vectors are then fed into the decoder, which typically reconstructs the output sequence via a cross-attention mechanism. A traditional training procedure includes a reconstruction loss between the decoder outputs and the initial inputs, as well as a quantization loss.

This traditional approach of relying on a single hidden vector can be insufficient for representing complex sequences like those in CAD. To obtain richer primitive representations, previous CAD works (Xu et al., 2022; 2023) separated CAD objects into multiple primitives and trained individual VQ models on them. We argue that this approach has limitations because training separate models can introduce alignment issues, and primitives may not receive contextualized information that could be beneficial for their encoding. In contrast, we split each CAD sequence into its constituent sketch-extrusion pairs and propose a primitive-specific pooling layer—replacing standard global pooling—to generate distinct representations for these pairs.

Our primitive pooling is achieved by masking; i.e., we generate masks for each input sketch-extrusion pair to isolate each primitive within it. Figure 8 demonstrates our method. In detail, we extract pooling masks from the input sequence that reflect the primitive corresponding to each token. Subsequently, we perform element-wise multiplication between each individual mask and the encoded representation output by the encoder. At this point, the resulting matrix contains information related only to the specific primitive isolated by the mask, allowing us to perform max pooling to obtain the pooled representation for that primitive. By concatenating these pooled primitive representations, we obtain a sequence of primitive-specific representations used for the decoder's cross-attention mechanism and for subsequent LLM alignment.

We set the maximum number of primitives in our method to 12. This corresponds to 9 flexible slots for loop primitives and 3 fixed slots for extrusion primitives. The limit for loop primitives is generally not restrictive, as the average number of loops per sketch-extrusion pair in our dataset is approximately 4, and during VQ-VAE training, we filtered out only around 900 sequences (out of 90,000) due to exceeding this limit.

## B.2 FSA SAMPLING

Our FSA sampling is based on a specially-designed Finite State Automaton (FSA) that determines the set of permissible next tokens (referred to as logit options). The system generates a token (or tokens) based on these allowed options, and then updates the FSA by inputting the current FSA state and the last generated token as an action, to determine the options for the subsequent step.

The algorithm is described in Algorithm 1, and the FSA's detailed design is illustrated in Figure 3. The specific logit masks employed by the FSA are listed in Table 6.

| Mask Name | Mask Content | Description |
|---|---|---|
| Init | [line, arc, circle] | Enforces that the first token is a curve type. |
| Numbers | [0-63] | Restricts selection to numerical values (0-63). |
| End-of-curve | [<curve_end>] | Enforces the <curve_end> token. |
| Primitive-start | [line, arc, circle] | Enforces a new curve; identical to Init. |
| End-of-loop | [<loop_end>] | Enforces the <loop_end> token. |
| End-of-face | [<face_end>] | Enforces the <face_end> token. |
| End-of-sketch | [<sketch_end>] | Enforces the <sketch_end> token. |
| B | [add, cut, intersect] | Boolean extrusion operations. |
| V | [0-63] | Numerical values for V parameter. |
| T | [0-63] | Numerical values for T parameters. |
| R | [-1, 0, 1] | -1, 0, or 1 for rotation parameters. |
| S | [0-63] | Numerical values for the S parameter. |
| O | [0-63] | Numerical values for O parameters. |
| End-of-extrusion | [<extrusion_end>] | Enforces <extrusion_end>. |
| Pad | [pad] | Restricts selection to the padding token. |

Table 6: Details of the logit masks provided by the FSA to guide token generation during decoding.

| Methods | COV | JSD | MMD | VLM(left) |
|---|---|---|---|---|
| **GPT-4o <–> Original Seq.** | 60.8 | 6.75 | 2.39 | 1.94 |
| **CAD-Tokenizer <–> Original Seq.** | 52.4 | 2.54 | 1.97 | 5.09 |
| **Original Seq. <–> Ground Truth** | 55.6 | 6.07 | 2.54 | 2.02 |

Table 7: Distributional measures for pairwise sequence comparisons from the CAD editing task. VLM scores of the left-hand-side component are reported.

## C EXPERIMENTAL RESULTS

### C.1 EVALUATION SETUP

**Instruction of Human Judges.** Five human judges were provided with 50 generation outputs for evaluation (40 from editing tasks and 10 from Text-to-CAD tasks). All participants had completed college-level to graduate-level education. The judges were given specific instructions on the evaluation task and were shown visual examples similar to those in the main qualitative results, but without model identifiers.

```
1 """
2 The following are a series of pictures. The upper half is the instruction
      and the standand answer. You need to rank the following three
      pictures based on their response quality to the instruction.
3
4 For example, if in a picture you like 3 the most, 1 following, and 2 the
      worst, rank them as (2 3 1).
5 """
```

Listing 1: Instructions of human judges.

**Instruction of VLM.** The prompts we use to generate VLM scorings are listed Listing 3 and 4.

```
1 """
2 The following is an original image of a CAD instance, a text description
      on editing and an image of the edited result. Measure if the figure
      corresponds to the given description, and give a score in the scale
```

```
      of 10. Only return the score. Do not comment on issues such as
      texture, smoothness and colors.\n description:{description}\n
3 """
```

Listing 2: Instructions of VLM on CAD Editing

```
1 """
2 The following is a text description of a 3D CAD figure and an image of a
      CAD instance. Measure if the figure corresponds to the given
      description, and give a score in the scale of 10. Only return the
      score. Do not comment on issues such as texture, smoothness and
      colors.\n description:{description}\n
3 """
```

Listing 3: Instructions of VLM on Text-to-CAD generation

## C.2 EXPERIMENTAL RESULTS

### C.2.1 ELABORATION ON CAD EDITING RESULTS ON THE DISTRIBUTIONAL METRICS

In the CAD Editing task, we observe that models such as GPT-4o can achieve high scores on distributional metrics for the CAD editing task, despite poor actual performance in editing. We hypothesize that a model can frequently fail to apply edits but still achieve satisfactory distributional scores by simply reproducing the input CAD sequence without modifications. To investigate this matter further, we provide additional distributional measures for three comparison types: 1. between GPT-4o outputs and the original input sequences, 2. between outputs from CAD-Tokenizer and the original input sequences, and 3. between the original input sequences and the ground truth edited sequences.

The results are displayed in Table 7. As shown in the third row (comparing original inputs to the ground truth), the original sequence yields high distributional scores. However, such outputs should be considered failed edits, as they do not incorporate any of the specified editing instructions. VLMs capture instruction-following capabilities much better by assigning a low score when an output merely returns the original sequence. In light of this, we measured the distributional similarity between the original input sequences and the outputs from GPT-4o and our CAD-Tokenizer. It was found that GPT-4o's outputs are much closer to the original sequences than those from CAD-Tokenizer, suggesting that our model more consistently attempts to modify the original sequence and perform the requested edits. This pattern, along with our model's overall advantages, is more accurately reflected by the difference in VLM scores presented in the main quantitative results.

We draw the conclusion that the interpretation of the distributional metrics should be extra careful as it may not always reflect the true performance of the model. The analysis on the CAD Editing performance should take all metrics (especially the qualitative ones) into consideration in order to get unbiased information.

### C.2.2 ADDITIONAL QUALITATIVE RESULTS

We present additional qualitative results for CAD editing and Text-to-CAD generation in Figure 9 and 10, respectively.

### C.2.3 FAILURE CASES

In this section, we discuss failure cases observed with our method. As shown in Figure 11, we classify these failures into three categories: *overcomplicated testing entries*, *misleading instructions*, and *lack of spatial or commonsense reasoning*. We list our findings below:

1. For the first category, the model faces difficulty in generating correct shapes from complicated instructions. For example, an extrusion resembling the number '4' poses difficulty, especially when the instruction phrases it as "four polyhedra" without stating its resemble with the number.

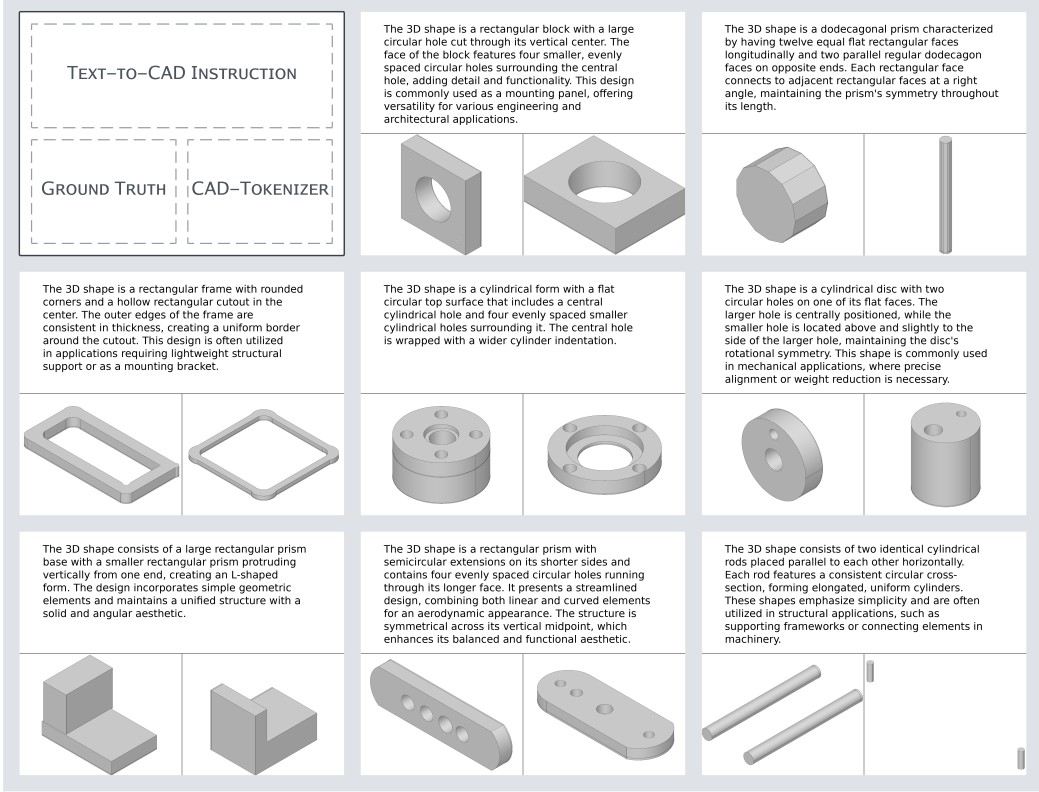

Figure 9: Additional qualitative results for CAD editing.

2. The second category describes instances where the instructions do not always accurately reflect the relationship between the original and target shape. This issue can arise because the CAD-Editor dataset was generated by prompting VLMs, for which achieving absolute robustness and trustworthiness remains an open problem; this, in turn, poses instruction-following challenges for our model.

3. The last category reflects a lack of common sense or spatial reasoning, likely originating from limitations in the backbone LLM's capabilities. Generating concrete shapes such as a "key" or letter shapes, as well as arranging multiple objects in an organized way, remains challenging.

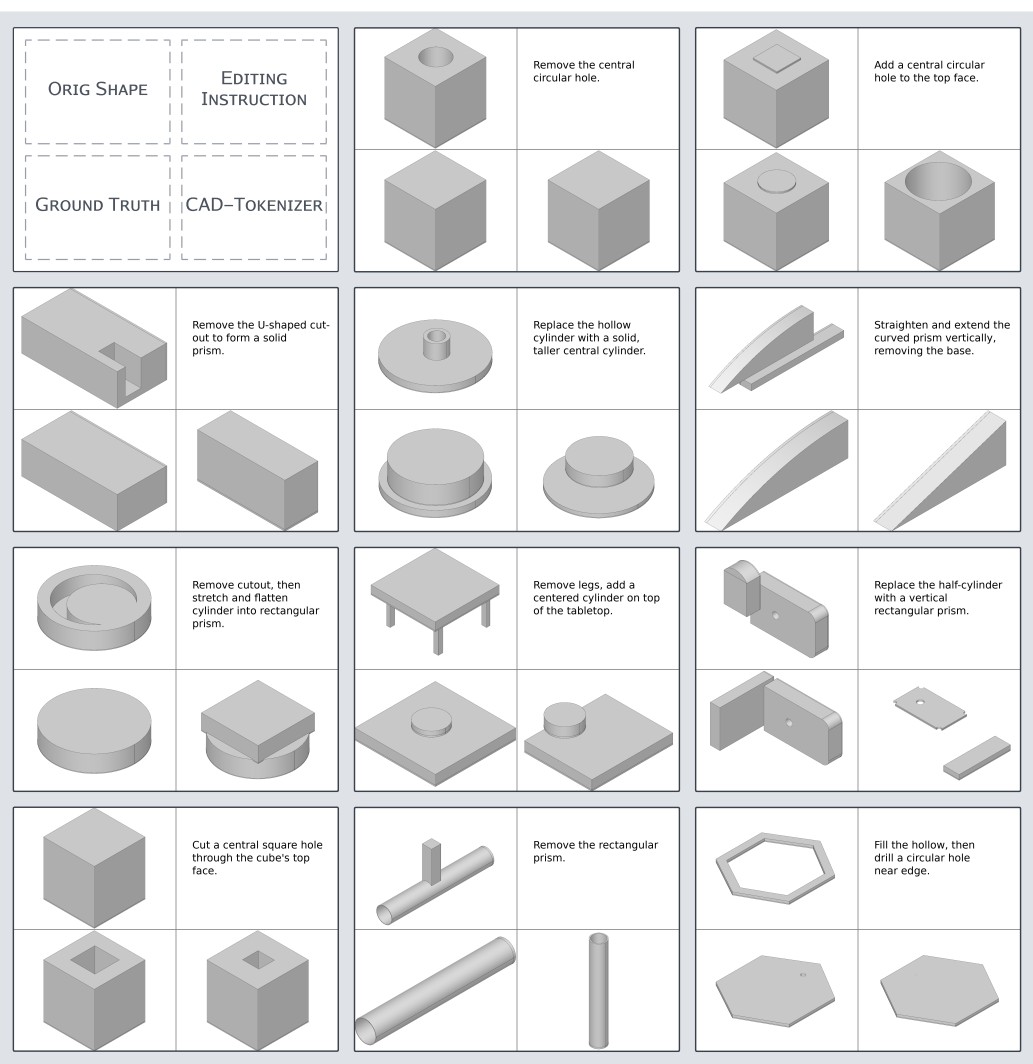

Figure 10: Additional qualitative results for Text-to-CAD generation.

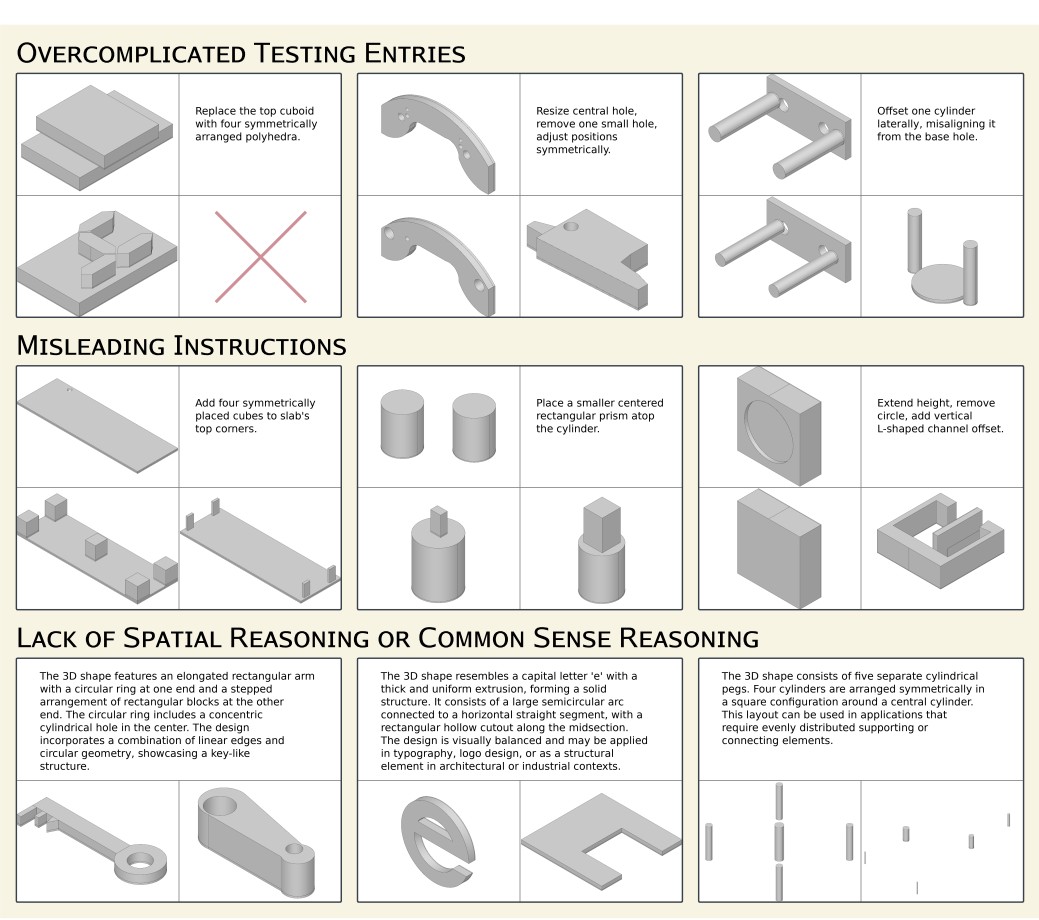

Figure 11: An overview of our model's failure cases. The layout follows that of Figures 9 and 10.

