# OpenReview forum: "CAD-Tokenizer: Towards Text-Based CAD Prototyping via Modality-Specific Tokenization"
_ICLR.cc/2026/Conference — ICLR 2026 Poster_

### Official Review · Reviewer_9hkP · 2025-10-23

**Soundness:** 3
**Presentation:** 2
**Contribution:** 3
**Rating:** 4
**Confidence:** 4

**Summary:**

The paper introduces a novel framework for unified text-guided CAD prototyping that jointly handles both Text-to-CAD generation and CAD editing.

The authors propose a primitive-level VQ-VAE pooling strategy to compress sketch–extrusion pairs into discrete, compact and semantically meaningful tokens.

These tokens are aligned with the LLM’s embedding space via lightweight adapters, enabling efficient fine-tuning.

Additionally, a finite-state automaton(FSA) guides decoding to enforce syntactic validity.

Experiments show that CAD-Tokenizer compresses and reconstructs sequences well across quantitative metrics. Ablation study proves the effectiveness of designed FSA-based sampling.

**Strengths:**

1. The proposed primitive-specific pooling layer is thoughtfully designed and its effectiveness is convincingly demonstrated through the reconstruction experiments reported in Table 2.

2. The evaluation across different levels of primitives—curves, loops, and single primitives—in Table 2 and Figure 4 clearly illustrates the trade-off between reconstruction quality and compression rate, making this key insight readily interpretable.

3. The utility of the finite-state automaton (FSA) is well supported by the ablation study in Table 4, which compellingly demonstrates its contribution to overall performance.

**Weaknesses:**

While the technical contribution of the paper is sound, the writing suffers from several issues that hinder readability and clarity:

Major writing weaknesses:

1. The formula at Lines 191-194 is really confusing.

Several notational and conceptual ambiguities:

- The term $\mathbf{l}_i$ on the right-hand side of the EMD is not explicitly defined; while it likely represents a target distribution or point set at step $i$ in a sequential modeling context, this assumption requires clarification.

- The notation { $p'$ } (set-bracketed) is weird, which are typically indexed without set notation. And its subscript seems to be missing.

- The definition of $E$ (the superscript of $T$) is unclear.

- What is $t_{1,i-1}$?

2. The reviewer recognizes that the main technical contributions of the paper are the primitive-specific pooling layer and FSA-based sampling, supported by thorough experiments on these components.

However, the details of the proposed primitive-specific pooling are entirely placed in the Appendix. The reviewer believes that this component should be elaborated in the main body of the paper.

> Meanwhile, the Abstract and Introduction emphasize the main contribution as the unification of Text-to-CAD generation and CAD editing. The reviewer does not observe an obvious technical novelty that directly enables this unification. (Notably, the primitive-specific pooling layer demonstrates its value by enabling a trade-off between reconstruction quality and compression rate, while FSA-based sampling appears to significantly reduce the Invalidity Ratio.)

Minor writing weaknesses:

1. The abbreviation “FSA” is introduced in line 156 without prior definition, which may confuse readers encountering the term for the first time.

2. Lines 88–89 introduce the finite-state automaton (FSA) for the first time, yet it is neither mentioned in the abstract nor in any preceding section.

3. Some recent works on CAD generation are missing from the related work section. LLM-based methods are evolving rapidly, including a more comprehensive coverage of recent work would strengthen the paper’s survey:

- CAD-MLLM: Unifying Multimodality-Conditioned CAD Generation With MLLM, Xu et al. (https://arxiv.org/abs/2411.04954)

- CAD-GPT: Synthesising CAD Construction Sequence with Spatial Reasoning-Enhanced Multimodal LLMs, Wang et al. (https://arxiv.org/pdf/2412.19663)

- CAD-Coder: Text-to-CAD Generation with Chain-of-Thought and Geometric Reward, Guan et al. (https://arxiv.org/abs/2505.19713)

- CADmium: Fine-Tuning Code Language Models for Text-Driven Sequential CAD Design, Govindarajan et al. (https://arxiv.org/abs/2507.09792)

**Questions:**

The reviewer acknowledges the technical contribution of primitive-specific pooling layer and FSA-based sampling.

However, if the authors insist that the main focus of the paper is the “unification of Text-to-CAD generation and CAD editing”, the reviewer would have some essential concerns/questions:

1. Are the primitive-level VQ-VAE tokenizer and FSA-based sampling genuinely helpful for the **CAD-Editing** task? If so, how can this be demonstrated through ablation studies?

2. Is the "unification of Text-to-CAD generation and CAD editing" beneficial to both **CAD-Editing** and **Text-to-CAD generation** if we exclude the improvements introduced by the primitive-level VQ-VAE tokenizer and FSA-based sampling?  How to prove it through experiment?

Some additional important questions:

1. In Figure 2(4), why is the instruction fed into the Encoder+Adapter? In contrast, in Figure 2(1), only the CAD sequence is fed into the encoder.

2. Why does Figure 2(1) include “PRIM. TOKENS(RAW)”? It appears that both the VQ latent tokens and raw primitive tokens are input to the decoder for sequence reconstruction—can this be clarified?

3. Why does Figure 2(2) also include “PRIM. TOKENS(RAW)”? What does the arrow labeled “For Training” signify?

4. Although FSA-based sampling's ablation study is convincing, the reviewer is curious about why IR is not an exact 0% because invalid sequence may not be sampled due to masking in Figure 8 from Appendix.

---

> ### Author Response · Authors · 2025-11-18
> **Response to Reviewer 9hkP (1/2)**
>
> We thank the reviewer for the valuable feedback and appreciate the opportunity to clarify the raised points.
>
> ## 1. Writing Weaknesses
>
> We thank the reviewer for the helpful suggestions. We have **updated the manuscript** to improve clarity and readability.
>
> Regarding the formula in Lines 191–194, we have revised the corresponding explanation with more descriptions of each component. Here we **restate the details** for your clarity.
>
> The tokenizer is optimized with two types of losses: a **reconstruction loss** and a **VQ loss**. The **reconstruction loss** is autoregressive, as the decoder is a Transformer module. For a sequence of length $n$, we sample the decoder with (1) the sequence prefix ($t\_{1, t-1}$) and (2) the primitive tokens ${p'}$, where the full form is $\\{p'\_1, \\ldots, p'\_k\\}$. The predicted distribution at index $i$ is compared against the ground-truth one-hot encoding $\textbf{1}\_i$.
>
> The **VQ loss** is accumulated rather than computed once per sequence, differing from traditional setups where a single quantized token is produced per run. Specifically, for a primitive token at index $j$, the pre-quantization vector is $\\overline{T}^E\_j$ and the quantized output is $p\_j$. Here, $T^E \= \\text{Encoder}(T)$ and $\\overline{T}^E \= \\text{Pooling}(T^E)$.
>
> We hope this clarification helps the reviewer better understand our formulation.
>
> Additionally, we have moved the FSA-related algorithm and figure from the appendix into the main text (previously omitted due to ICLR’s page-limit restrictions) and updated the Related Work section to include the most recent literature.
>
> ## 2. Importance of the Unification of Editing and Text-to-CAD
>
> We would like to clarify that the "*unification of Text-to-CAD generation and CAD Editing*" is **not mutually beneficial**. Its significance falls in enabling a single framework applicable to **industrial pipelines**. This unified setup poses much higher requirements on the backbone model: text-to-CAD generation demands strong generation capability, while CAD editing emphasizes strict instruction following—two distinct skill dimensions.
>
> As shown in Table 1, our vanilla-LLaMA experiment (without the VQ-VAE tokenizer or FSA-based sampling) demonstrates that naively fine-tuning an LLaMA model on the unified task **significantly degrades** performance on both subtasks. Our proposed tokenization algorithm alleviates this issue by reducing the backbone’s burden, allowing the LLM to focus on modeling inter-primitive relationships. The accompanying pooling and sampling mechanisms are designed to support this unified formulation.
>
> ## 3. The Helpfulness of the Technical Contributions on CAD Editing
>
> Below are the CAD editing results comparing our full model with ablations that remove specific components:
>
> | Method | F1-Avg | CD | COV | JSD | IR |
> | :---- | :---- | :---- | :---- | :---- | :---- |
> | Ours | 91.7 | 13.5 | 52.4 | 2.54 | 8.38 |
> | Ours w/o primitive pooling | 80.4 | 52.4 | 56.5 | 44.7 | 72.8 |
> | Ours w/o FSA sampling | 82.9 | 58.9 | 34.5 | 57.7 | 14.4 |
>
> The substantial performance drop in both ablations confirms that primitive pooling and FSA sampling are crucial to model effectiveness on Editing.
>
> ## 4. The Helpfulness of Unification on CAD Editing and Text-to-CAD
>
> Without the proposed technical contributions, unification provides **no benefit** for either task and even **harms performance**. As discussed earlier, the vanilla-LLaMA baseline reflects such a unified setting by being trained jointly on both tasks, yet it exhibits a large performance gap compared to our framework (see Table 1(a)(b), rows 3 and 4). This underscores the necessity of our tokenizer and sampling design for effective unification.

---

> > ### Author Response · Authors · 2025-11-18
> > **Response to Reviewer 9hkP (2/2)**
> >
> > ## 5. Answers to the Additional Questions
> >
> > Before addressing each individual question, we would like to **briefly elaborate** on the demonstration in Figure 2\. Figure 2(1) illustrates our VQ pretraining stage, where we train a VQ-VAE module with primitive pooling to produce raw, primitive-specific tokens. Figure 2(2) shows how these tokens are further aligned to the LLM embedding space using an additional pair of adapters. After this stage, the VQ tokens become eligible for training; all CAD sequences are translated into these tokens for finetuning the backbone LLMs (Figure 2(3)). Figure 2(4) provides an overview of the inference process.
> >
> > Primitive tokens generated by our system are denoted as `raw` if they come from a pre-trained VQ-VAE that has *not* been LLM-aligned (i.e., the model only completed the procedure in Figure 2(1)). They become `final` if the VQ-VAE includes the trained, LLM-aligned adapters (the stage shown in Figure 2(2)).
> >
> > 1. As shown in Figure 2(2), after pretraining the VQ-VAE, an additional alignment stage introduces two lightweight MLP adapters to the VQ module. These adapters project token embeddings into the LLM’s embedding space so that the backbone can recognize and utilize them. The full process involves encoding the CAD sequence, then passing it through the adapters before feeding it to the LLM.
> > 2. Figure 2(1) illustrates pre-alignment tokens (PRIM. TOKENS \[RAW\]), which are not yet interpretable by the LLM. In Figure 2(2), the corresponding tokens are annotated as PRIM. TOKENS \[FINAL\], indicating they have passed the alignment adapters.
> > 3. See our answer above; Figure 2(2) replaces RAW tokens with FINAL ones due to this alignment stage. These tokens are then passed to the backbone LLM for fine-tuning.
> > 4. The invalidity ratio reflects two aspects: **grammatical correctness** and **geometric renderability**. However, a CAD sequence can be both grammatically correct and geometrically unrenderable. For example, one could draw two valid circles with edges that collide, forming an invalid face. While we do eliminate grammatical issues and observe a significant decrease in geometric issues, the latter is still allowed to occur, which accounts for all cases that make our invalidity ratio non-zero.

---

> > > ### Comment · Reviewer_9hkP · 2025-11-19
> > >
> > > The reviewer acknowledges the author's reply. The new ablation table is helpful to illustrate the necessity of the proposed module. Please don't forget to include it in the main paper or the appendix in the later revision to improve the paper.
> > >
> > > I have some more questions that are not fully clear to me, and I hope to get them clarified.
> > >
> > > 1. Since the author clarifies that:
> > >
> > > > Primitive tokens generated by our system are denoted as `raw` if they come from a pre-trained VQ-VAE that has not been LLM-aligned (i.e., the model only completed the procedure in Figure 2(1)). They become `final` if the VQ-VAE includes the trained, LLM-aligned adapters (the stage shown in Figure 2(2)).
> > >
> > > From Figure2(1) and Figure2(2), it seems that the input of `Transformer encoder` in (1) and `VQ Encoder` in (2) is the same. If `raw` is not used, why do we need to get `raw`? Why don't we directly obtain `final`?
> > >
> > > 2. In Table3, why is `CAD-Tokenizer single` as bad as 70% invalid rate? 70% would indicate that this setup totally doesn't work. In the current version, the paper describes that:
> > >
> > > > CAD-Tokenizer (single), which omits primitive pooling and encodes each sketch–extrusion pair as a single unit.
> > >
> > > It's not so much reasonable to me why simply without primitive pooling would make everything that bad. (A similar conclusion can be observed from your new attached table in reply to me.)

---

> > > > ### Author Response · Authors · 2025-11-20
> > > > **Response to the Additional Questions**
> > > >
> > > > We thank the reviewer for the acknowledgment and the helpful suggestion regarding the ablation table. We will ensure this table is included in the final revision.
> > > >
> > > > ## 1. The Two-Stage VQ paradigm
> > > > First, regarding the architecture shown in Figure 2(1), we define the `VQ Encoder` as the composition of the `Transformer Encoder`, `Primitive Pooling`, and `VQ Nearest Retrieval` modules at the bottom of that subfigure. For a clearer visualization of the VQ pretraining process, we refer you to Figure 7, which illustrates this flow in greater detail.
> > > >
> > > > Second, the `raw` tokens are not required for subsequent inference steps. We visualize them here for two reasons: 1\) to explicitly demonstrate the correspondence between raw tokens and input primitives (indicated by color), and 2\) to highlight that the alignment procedure in Figure 2(2) is specifically designed to optimize the projection between raw and final tokens. This alignment depends on accessing the raw token production during the training phase.
> > > >
> > > > Third, we do not train this two-stage process jointly because the objectives differ fundamentally. The VQ modules are required to learn the structural patterns of CAD primitives, whereas the adapter layers are responsible solely for projecting between the two token spaces. Joint training creates instability, as the adapter layers would struggle to converge without receiving stabilized primitive representations. Therefore, decoupling these procedures is a practical necessity for stable training.
> > > >
> > > > ## 2. Bad CAD-Tokenizer Single Results
> > > > The poor performance of this variant stems from severe information loss caused by over-compression. To illustrate this, consider the following example sequence containing $2$ loops with $n+m$ curves:
> > > >
> > > > ```
> > > > [curve parameters] <curve_end> (n curves in this loop) <loop_end> [curve parameters] <curve_end> (m curves in this loop) <loop_end><face_end><sketch_end> [extrusion parameters] <extrusion_end>
> > > > ```
> > > >
> > > > Our default tokenizer (curve-level) assigns a primitive token to each curve, resulting in a total of $n+m+5$ tokens (the extrusion parameters constantly produce 5 tokens):
> > > >
> > > > ```
> > > > |curve-token| |curve-token|, …, |curve-token| |extrusion-token-1|, …, |extrusion-token-5|
> > > > ```
> > > >
> > > > The (loop) variant operates at a looser granularity, reducing the sequence to $2+5=7$ tokens:
> > > >
> > > > ```
> > > > |loop-token| |loop-token| |extrusion-token-1|, …, |extrusion-token-5|
> > > > ```
> > > >
> > > > In contrast, the (single) variant attempts to encode the entire sequence into a **single** token:
> > > >
> > > > ```
> > > > |sketch-extrusion-token|
> > > > ```
> > > >
> > > > As shown, this approach is excessively compressive, making it impossible to retain the necessary geometric details of the original sequence.

---

> ### Comment · Reviewer_9hkP · 2025-11-20
>
> The reviewer thanks the authors for the clear explanations. I have raised my score since my primary concerns have been addressed.
>
> Additional suggestions:
>
> ### 1. Regarding minor writing weakness #3 mentioned in my initial review:
>
> In line 101, it appears that Cad-MLLM is cited as being conditioned on hand sketches is not accurate, based on my knowledge.
>
> Similarly, the reference to the Text-to-CAD task in lines 114–115 may not be entirely precise. The authors might consider describing and discussing it as a multimodal method, and possibly adding Cadrille alongside it, since it is also multimodal. (But you may consider separating command sequence based(CAD-MLLM) and DSL code based(Cadrille) when discussing in related work for a better clarification)
>
> ### 2. Quantitative comparison
>
> In Figure 5, the reviewer finds it difficult to clearly observe the superior qualitative performance from the selected case. This may weaken the argument for the proposed methodology. However, the reviewer acknowledges that the overall quantitative results effectively demonstrate the method’s advantages. The authors may consider selecting more distinct and challenging examples to better highlight the qualitative improvements.

---

> > ### Author Response · Authors · 2025-11-21
> > **Thank You**
> >
> > We thank the reviewer for your recognition of our work. We will update the relevant sections in the next version of our manuscript.

---

### Official Review · Reviewer_GPRz · 2025-10-26

**Soundness:** 3
**Presentation:** 3
**Contribution:** 3
**Rating:** 4
**Confidence:** 3

**Summary:**

This paper presents a unified LLM-based model for text-driven generation and editing of Computer-Aided Design (CAD) models. The authors design a CAD VQ-VAE that compresses CAD operations into compact representations, which can tokenize CAD sequences into discrete codes. In addition, they also introduce linear adapters that align VQ embeddings with the LLM through an additional training phase. Experimental results demonstrate that the proposed method achieves state-of-the-art performance.

**Strengths:**

The paper demonstrates thoughtful design of the CAD tokenizer to enable seamless integration with the LLM framework, effectively bridging the gap between continuous CAD representations and discrete language model inputs.

**Weaknesses:**

1. What are the typical lengths of CAD sequences in your dataset? I have concerns about the scalability of the proposed pipeline. While the transformer encoder can handle variable-length inputs and capture relational information, the primitive pooling operation fuses all tokens within each primitive into a single latent vector before VQ encoding. As CAD sequences grow longer, the number of operation types remains constant, but the complexity of inter-operation relationships increases significantly. It is unclear whether this pooling-then-quantizing approach can adequately capture these increasingly complex relationships in longer sequences.
2. Each extrusion operation contains precise parameters such as translation and rotation values. However, the VQ tokenizer maps continuous embeddings to a finite discrete codebook, which may result in quantization loss. This raises concerns about whether fine-grained geometric details are preserved, potentially leading to inaccuracies in the reconstructed parameter values.

**Questions:**

Please refer to the weaknesses.

---

> ### Author Response · Authors · 2025-11-18
> **Response to Reviewer GPRz**
>
> We thank the reviewer for the constructive feedback and appreciate the opportunity to address these concerns.
>
> ## 1. Scalability
> Before preprocessing, each CAD sequence contains 1-68 sketch-extrusion pairs (S-E pairs for simplicity). We clip the abnormally long sequences so that only the ones with less than 20 S-E pairs are selected for training. In each S-E pair, there are on average 28.9 curves. To ensure manageable token lengths, we filter out any sketch–extrusion pairs with more than 59 loops, limiting each pair to at most 64 tokens (since each extrusion is represented by 5 tokens).
>
> We realize that the reviewer’s comment on this part may stem from a **misunderstanding** of our paper. We form all tokens representing **each individual primitive** into a single latent vector, **not all tokens within each primitive category**. For example, if a CAD sequence originally consists of 3 circles, 2 lines, and an arc, we tokenize it into $3+2+1=6$ tokens rather than $1+1+1$ tokens (i.e. merging all 3 circles into one token, etc.). When the sequence grows longer, the length of our encoded sequence increases as well.
>
> Therefore, regarding scalability, our tokenizer addresses this issue more effectively than the default text tokenizer. In fact, its advantage increases with longer sequences. This is because our primitive pooling and overall design explicitly leverage the VQ-VAE’s capability to capture **primitive-level features**, enabling the backbone LLM to better model **inter-primitive relationships** (see Line 76). As shown by table 1, the LLM’s performance would degrade significantly without our proposed setup.
>
> ## 2. Quantization Loss
>
> First, using discrete codebooks to process continuous vectors **is a common technique** in generative domains. As long as the reconstruction error is reasonably controlled, the downstream model typically learns more effectively from the discrete data compared to continuous data. This approach applies not only to image and video generation [1-3] but also to early-stage CAD generation [4-5].
>
> Second, the design we proposed is intended to preserve the geometric details in the extrusion part. As detailed in Appendix A.2 (around Line 782), our primitive definition splits each extrusion sequence into three parts, represented by five tokens, each corresponding to specific operational parameters.
>
> Below, we compare the reconstruction quality of encoding the extrusion as one token versus five tokens (our setup):
>
> | Methods | F1-Skt | COV | JSD | Compression rate(\%) |
> | ---- | ---- | ---- | ---- | ---- |
> | CAD-Tokenizer-5tok | 94.1 | 64.5 | 8.19 | 21.4 |
> | CAD-Tokenizer-1tok | 81.1 | 59.9 | 40.5 | 16.7 |
>
> This comparison demonstrates that using more fine-grained, five-token representations to encode extrusions substantially improves the preservation of geometric details compared to the single-token variant, and reduces the quantization loss as you proposed.
>
> Third, a significant improvement would be to use a continuous VAE as token generators for generative learning. Discoveries in other domains, such as image generation [6], support this direction, and we recommend it for future work.
>
> [1] Zero-Shot Text-to-Image Generation. https://arxiv.org/abs/2102.12092
>
> [2] MAGVIT: Masked Generative Video Transformer. https://arxiv.org/abs/2212.05199
>
> [3] OmniTokenizer: A Joint Image-Video Tokenizer for Visual Generation. https://arxiv.org/abs/2406.09399
>
> [4] SkexGen: Autoregressive Generation of CAD Construction Sequences with Disentangled Codebooks. https://arxiv.org/abs/2207.04632
>
> [5] Hierarchical Neural Coding for Controllable CAD Model Generation. https://arxiv.org/abs/2307.00149
>
> [6] Transfusion: Predict the Next Token and Diffuse Images with One Multi-Modal Model. https://arxiv.org/abs/2408.11039

---

> > ### Comment · Reviewer_GPRz · 2025-11-24
> >
> > Thanks for the authors' response. I have increased the score. Additionally, please double check Table 1. There are some labeling mistakes.

---

> > > ### Author Response · Authors · 2025-11-24
> > > **Thank You**
> > >
> > > Thank you for your recognition of our work. We will update the table in the next version of our manuscript.

---

### Official Review · Reviewer_A1Ms · 2025-10-31

**Soundness:** 3
**Presentation:** 3
**Contribution:** 3
**Rating:** 8
**Confidence:** 3

**Summary:**

This paper introduces CAD-Tokenizer, a unified framework for text-based CAD prototyping that integrates both Text-to-CAD generation and CAD editing within a single model. The authors argue this joint formulation better reflects real industrial design workflows that are inherently iterative.

The core contribution is a tokenization method that replaces standard LLM tokenizers. Using a VQ-VAE, CAD operations are converted into discrete primitive-level tokens, yielding representations aligned with the structural and semantic nature of CAD data. An LLM is fine-tuned on CAD tokens to perform both generation and editing. This joint training improves performance and computational efficiency.Finite-State Automaton is used during decoding to enforce syntactic validity, ensuring outputs adhere to the CAD grammar. Experiments show CAD-Tokenizer outperforms task-specific baselines (CADFusion, CAD-Editor) and general models like GPT-4o across diverse metrics, including geometry, instruction following, and human evaluation.

**Strengths:**

Unifying CAD generation and editing is an important conceptual step that aligns research with practical design workflows.

The insight that standard tokenizers fail to capture CAD’s structured semantics is compelling, and the proposed primitive-aware tokenizer addresses this gap effectively.

The combination of VQ-VAE tokenization, adapter-based embedding alignment, and FSA-guided decoding is well-justified. Ablation studies clearly demonstrate each component’s benefit.

The experimental design is rigorous, using rich metrics and strong baselines. Results show consistent and convincing gains.

**Weaknesses:**

Due to limited open-source CAD datasets, experiments focus on simple shapes. The model struggles with complex geometries requiring advanced spatial reasoning.

Standard distributional metrics (COV, JSD) are shown to be unreliable for editing tasks, as they can reward unchanged outputs. Although alternative VLM and human metrics mitigate this, the issue underscores a broader challenge in evaluation.

Additionally, some errors stem from the LLaMA-3-8b model’s limited spatial reasoning, capping overall performance.

**Questions:**

- How does the tokenizer’s 2048-codebook design scale with increased CAD complexity? Would a larger codebook affect alignment or fine-tuning efficiency?
- How adaptable is CAD-Tokenizer to other CAD formats beyond Sketch-and-Extrude Modeling? Would retraining be necessary or could components be reused?
- What types of geometric errors (e.g., self-intersections, open sketches) persist despite FSA-based enforcement?
- Can you provide concrete comparisons of training or inference time relative to Vanilla-LLaMA?

---

> ### Author Response · Authors · 2025-11-18
> **Response to Reviewer A1Ms**
>
> We thank the reviewer for recognizing our work. We provide the following clarifications and hope they address your concerns.
>
> ## 1. Codebook Size
>
> We have experimented with various codebook sizes ranging from `1024` to `8192`. In our exploration, we did not observe a positive correlation between codebook size and tokenizer performance.
>
> ## 2. CAD-Tokenizer’s Adaptability
>
> If the target CAD format follows a modeling paradigm homogeneous to the sketch-and-extrude process, our framework can be directly adapted once an appropriate intermediate conversion is implemented. The main components of our system can be reused under this setup. Currently, our tokenizer is compatible with objects represented in `.step` and `.stl` formats.
>
> However, if the target CAD format differs substantially from our assumptions, retraining would be necessary to ensure proper adaptation.
>
> ## 3. FSA on Geometric Errors
>
> FSA does not directly correct geometric errors. Rather, it prevents the generation of sequences that violate the grammatical constraints of CAD operations. Empirically, we observed that incorporating FSA reduces the frequency of self-intersections and collisions. Note that *open sketches*—as defined by our grammar—do not occur in our generated outputs.
>
> ## 4. Time Cost Related to Vanilla-LLaMA
>
> Training for 20 epochs on our dataset involves approximately 650,000 steps (due to data parallelism). The vanilla LLaMA baseline required about 120 hours, whereas our proposed method completed training in roughly 30 hours. Our method is faster than the baselines because we encode the CAD sequences to shorter encoded sequences with less tokens each. This makes our approach empirically faster than the vanilla setup during inference as well.

---

> > ### Comment · Reviewer_A1Ms · 2025-11-21
> >
> > Thank you for your response. Few questions:
> >
> > > We have experimented with various codebook sizes ranging from 1024 to 8192. In our exploration, we did not observe a positive correlation between codebook size and tokenizer performance.
> >
> > Does this mean that performance decreases increasing the codebook? Or simply is not impacting performance? Do you think  this is a data issue or related to the model?
> >
> > > FSA does not directly correct geometric errors. Rather, it prevents the generation of sequences that violate the grammatical constraints of CAD operations. Empirically, we observed that incorporating FSA reduces the frequency of self-intersections and collisions.
> >
> > I find this confusing. Is the FSA reducing the incidence of errors, or making such errors (grammar violations) not possible?
> > You write that FSA prevents grammar violations but in the next line you say that is reducing collisions. Aren't collisions grammar violations?

---

> > > ### Author Response · Authors · 2025-11-21
> > >
> > > We appreciate opportunity to further address your concerns.
> > >
> > > ## 1\. Codebook
> > >
> > > The codebook size is **not impacting performance**, either in a positive or negative way. We attribute this stability to our primitive pooling strategy, which segments CAD sequences into granular chunks and allows the model to capture underlying patterns more effectively.
> > >
> > > Regarding the relationship between codebook capacity and data quality, there are valid arguments on both sides. We agree that **a larger codebook could theoretically capture more complex shapes** if the dataset quality improves, but since complex shapes are eventually decomposed into standard primitives (e.g., lines, arcs), **our current codebook remains sufficient to represent most of them** without requiring expansion.
> > >
> > > ## 2\. FSA Sampling
> > >
> > > There are two distinct types of errors a CAD generation system can make: **grammatical errors and geometrical errors**. The former involves **violations of syntax rules**, such as defining a circle with two coordinates when four are required. The latter refers to scenarios where the sequence is grammatically correct but results in **invalid geometry**, such as self intersections or collisions as mentioned.
> > >
> > > FSA sampling **strictly eliminates** grammatical errors. Interestingly, while not the primary intent of this mechanism, we empirically observed that it also **reduces the frequency** of the geometrical errors.

---

> > > > ### Comment · Reviewer_A1Ms · 2025-11-24
> > > >
> > > > > We attribute this stability to our primitive pooling strategy, which segments CAD sequences into granular chunks and allows the model to capture underlying patterns more effectively.
> > > >
> > > > This makes sense. The smallest codebook you are using has enough capacity to represent the primitives. Initially I thought that performance went up/down without a trend.

---

> > > > > ### Author Response · Authors · 2025-11-24
> > > > >
> > > > > We are glad to know that our clarifications addressed your concerns. We would be happy to hear from you if you have additional questions.

---

### Official Review · Reviewer_uche · 2025-11-03

**Soundness:** 1
**Presentation:** 3
**Contribution:** 2
**Rating:** 2
**Confidence:** 5

**Summary:**

Authors proposed a new tokenizer to unify text-based CAD generation / editing. More specially they first train VQ-VAE to represent CAD sketch and extrude as CAD-specific DSL tokens. They then align those token embeddings with pretrained LLM tokens, effectively converting DSL tokens into LLM text indices. Then the base LLM is SFT on the new sequence of token indices. Authors demonstrate good results compared to using HNC-cad and the default BPE tokenizer.

**Strengths:**

(1) The proposed VQ-VAE approach with max pooling is more token efficient than HNC-CAD.

(2) Aligning CAD DSL to pretrained LLM embedding is interesting and works surprisingly well.

(3) Authors demonstrate better result using the new LLM aligned CAD tokenizer.

**Weaknesses:**

(1) Sketch seqence and extrude representation largely follows previous work. Authors should make this more obvious as opposed to categorying those relevant works as "classical cad tasks". The CAD primitive-level parsing is actually a more compressed verison of hnc-cad with masking and single VQ-VAE as opposed to multiples.

(2) The main comparison to hnc-cad is not sound. hnc-cad encodes deepcad data into a sequence of discrete codes. So the claim that it is not designed for LLM training is not fair. Anything that representat data as discrete indices can be used for LLM training. In fact an autoregressive Transformer was trained from scratch for hnc-cad. Perhaps authors should more clearly state that a pretrained LLM was not trained on those discrete codes in that paper. But it should be easy to train one for ablation study, which the author did not do.

(3) The main comparison to BPE tokenzier is not correct. Authors directly apply text-based tokenizer on the parsed sketch-and-extrude sequence, which is very abstract dsl and different from text. Some works might have did this but it is very obviously bad practice. In this case the performance could be bad because the compatiblity issue with data format and pretrained LLM model, not because of the BPE tokenizer. A more fair comparision is to convert CAD DSL into text like CADQuery, JSON, or python-like code. Then after finetuning a LLM or Code-LLM with default text tokenizer and test if the model understand CAD constructions. Actually there are already many works in this direction (cad-llama) and has proved that this works well with low invalid ratio. This should be much better than the 88% BPE invalid ratio reported in the paper.

(4) Failure to compare to more recent text-based CAD generation / editing works such as CAD-Llama, CAD-MLLM and etc. Text generation and editing visual results also look much more simplier than those in related works. It will be more convincing to show that taken out the default text encoder in those models and switching to the one from the paper also leads to improvements.

(5) Unclear if the alignment step is still necesary if the LLM vocabulary was expended to include the new CAD tokens, and LoRA finetuning uses larger rank and also update the embedding and logits layer.

Overall, when it comes to something as fundemental as tokenizer, I hope the authors can be more careful about the comparision and make sure it is not a model or training issue that affects the final result. Right now it lacks comparsion to text-based CAD generation works and some important settings are not sounds. I feel liket this will have a high impact on the community. If the claims are true it will basically disencourge future works to directly train on DSL or text / code data paired with default text tokenizer. This is a huge change and should be better validated.

**Questions:**

(1) Please make fair comparison to hnc-cad and espeically other text-based CAD generative models. Directly applying text tokenizer on abstract DSL is not a sensible comparison. BPE baseline should be considered in the setting when CAD DSL is converted to text or python code etc.

(2)  After finetuning, would the LLM still maintain its original text generation and reasoning capability? Have authors did any ablation study on this? This could potentially be a huge disadvantage for this approach compared to text-based CAD generation, since for them the grammar and code structured are aligned with the pretrained model.

(3) Visual editing results are very simple, much more simple than what I am seeing in other works using pretrained LLM. I highly suggest authors compare to those more recent methods (cad-llama, ca-mllm etc).

(4) Have authors tried different LoRA settings? Recent experiments show that when data distirbution is different from training, it is ususally beneficial to use larger rank and sometimes full training is even better.

---

> ### Author Response · Authors · 2025-11-18
> **Response to Reviewer uche (1/3)**
>
> We sincerely appreciate your constructive feedback. Your knowledgeable insights are beneficial to us. While we do not fully agree with several of your interpretations, we believe many of the concerns arise from misunderstandings of our paper. We are grateful for the opportunity to clarify them.
>
> ## 1. "CAD Primitive-level Parsing is Actually a More Compressed Version of hnc-cad"
>
> Thank you for this comment. We have revised the related work section to better reflect classical CAD contributions, and we highlighted the changes in red for clarity.
>
> However, we respectfully note that the characterization of our method as merely “a more compressed version of hnc-cad” is an **over-simplification**.
>
> First, our tokenization approach differs fundamentally from hnc-cad. Unlike hnc’s hierarchical representation, we restructure CAD objects into a fully sequential form, which both simplifies the encoding pipeline and enables us to learn CAD patterns with a *single* VQ module.
>
> Second, achieving a functional “compressed variant” of hnc is far from trivial. Table 2 presents several CAD-Tokenizer variants, including a *single-token* version that encodes each sketch–extrusion pair as one token—closely matching the type of compression implied in your comment. For convenience, we reproduce key scores below:
>
> | Model Variant | F1-sketch | Coverage | JSD |
> | :---- | :---- | :---- | :---- |
> | Ours | 94.1 | 64.5 | 8.19 |
> | *single* | 76.5 | 54.0 | 35.9 |
> | hnc-cad | 85.5 | 57.5 | 29.8 |
>
> Two observations follow:
>
> 1. A naive “compressed VQ-VAE” performs significantly worse than both our method and hnc, highlighting that surpassing hnc is inherently difficult.
> 2. Our VQ tokenizer outperforms hnc in both reconstruction and compression quality (see also Figure 4), demonstrating that our contribution on solely CAD reconstruction is technically substantive and non-trivial.
>
> ## 2. The “Main Comparison” to hnc-cad and BPE
>
> Before addressing specific concerns, we clarify that these experiments are **ablation studies**, not our main comparison. Our actual main comparison appears in Table 1, which evaluates a range of domain-specific and general-purpose models.
>
> ### 2.1 Comparison with hnc
>
> We believe there is a **misunderstanding** of the *purpose* of this comparison. Our work provides a sound comparison with hnc-cad by measuring the **reconstruction quality** of the tokenizers.
>
> We detailed this part in Section 4.3, starting from Line 418\. Reconstruction quality measures the difference between the original CAD shape and the one being passed through the CAD encoder and decoder. It is the central property for evaluating a tokenizer, as it directly reflects the noise introduced before LLM training (A model with better reconstruction quality generates less noisy tokens for the next stage). This is a *fair and aligned* comparison because reconstruction is one of the core training objectives of both hnc and our tokenizer. As the results show, hnc does not achieve competitive reconstruction quality; therefore, we did not proceed with downstream training using hnc tokens.
>
> ### 2.2 Comparison with BPE
>
> We believe this misunderstanding stems from some **confusion** regarding the definition of our BPE method.
>
> The BPE you described is **exactly what we have conducted** in our work. Your suggestion is to convert CAD sequences to a more text-like or code-like modality - this is what our **vanilla-LLaMA** baseline has done. In general, we convert each CAD object into a text-like sequence and use the default text tokenizer to process it. An example is shown below (you can find this sequence and descriptions of each component in Appendix A.1):
> ```
> line, 10, 7, <curve_end> … <loop_end> … <face_end> … <sketch_end> … <extrude_end>
> ```
> As you can see in Table 1 (our main comparison), this setup completely fails for text-based CAD prototyping.

---

> ### Author Response · Authors · 2025-11-18
> **Response to Reviewer uche (2/3)**
>
> ### 2.2 Comparison with BPE, Continued
>
> The BPE we conducted was a variant trained directly on the CAD data. We did not reuse the text BPE here. We included this experiment to provide a more comprehensive evaluation of our proposed method. The rationale of our work, as stated in the introduction and demonstrated in Figure 1, is that the default text tokenizer (used by CAD-Llama, CADFusion, and various other works) is not optimal for improving CAD understanding. This is because it converts the sequence into smaller word pieces that cause attention layers to focus on punctuation or partial tokens instead of meaningful primitives (See line 53). Let’s take the previous line as an example: the default tokenizer splits it as below:
> ```
> |line| |,| |10| |,| |7| |,| |<| |curve| |_end| |>| … |<| |loop| |_end| |>| …
> ```
> This tokenization strategy definitely adds a burden to the backbone's ability to understand the DSL tokens. We attempted to solve this problem with our CAD-Tokenizer method. Our current BPE setup aims to determine if we can improve generation quality by simply using a non-neural baseline (as VQ-VAE is a neural network which requires computation). Under this hypothesis, it is possible for BPE to provide an optimal CAD encoding with more sensible compressions. We anticipate that applying the BPE algorithm to the word pieces will allow tokens to be combined to form more meaningful chunks for the backbone to learn. Still using the previous example, the sequence is likely to be joined into:
> ```
> |line, 10, 7,| |<curve_end> <loop_end>| … |<extrude_end>|
> ```
> Our ablation study on this setup, however, shows that this does not happen in the end.
>
> ## 3. "Failure to Compare to More Recent Text-based CAD Generation/editing Works"
>
> This is a **mistake**. *CAD-LLaMA* is a CVPR 2025 paper, and *CAD-MLLM* was released on arXiv in 2024\. Our baseline methods - *CADFusion* (ICML 2025\) and *CAD-Editor* (ICML 2025) - are both **newer** than the mentioned works.
>
> Moreover, as described in the previous section, our vanilla-LLaMA setup **closely resembles** *CAD-LLaMA*. Both adopt the same tokenization procedure, backbone model, and fine-tuning paradigm (i.e., only SFT, without RL or other advanced techniques).
>
> ## 4. The Necessity of Alignment if the LLM Vocab is Expanded
>
> We would like to share an **additional observation** from the early stage of our exploration. We attempted to expand the LLM vocabulary with new CAD tokens. Specifically, LLaMA 3 provides 256 reserved tokens (IDs `128001`–`128256`), which we assigned to CAD-specific tokens. However, this experiment completely failed - producing 100\% invalid outputs - which led us to exclude it from the current manuscript.
>
> The model was unable to generate renderable CAD sequences. In particular, it frequently confused similar tokens (e.g., `<|reserved_special_token_19|>` vs. `<|reserved_special_token_18|>`), making the results entirely unparseable.
>
> While we acknowledge that such a practice may succeed in other domain-specific LM scenarios, we respectfully note that the CAD domain is strongly limited by data availability, as also recognized by Reviewer `A1Ms`.

---

> ### Author Response · Authors · 2025-11-18
> **Response to Reviewer uche (3/3)**
>
> ## 5. LLM’s Text and Reasoning Capability
>
> We did not observe a significant loss in text or reasoning capability during inference. However, since previous CAD models have not evaluated such abilities after fine-tuning, there is no established protocol for comparison.
>
> To illustrate our findings, we conducted several **qualitative question-answering tests** to evaluate factual reasoning, reading comprehension, and writing ability. Example interactions are shown below:
>
> ```
> User: The capital of France is
> Model: Paris
>
> User: Who was the president of the United States during WWII?
> Model: Franklin Roosevelt was the president during WWII.
>
> User: Answer the question after reading the excerpt: The Airbus A340 is a long-range, wide-body passenger airliner that was developed and produced by Airbus. In the mid-1970s, Airbus conceived several derivatives of the A300, its first airliner, and developed the A340 quadjet in parallel with the A330 twinjet. In June 1987, Airbus launched both designs with their first orders, and the A340-300 took its maiden flight on 25 October 1991\. It was certified along with the A340-200 on 22 December 1992 and both versions entered service in March 1993 with launch customers Lufthansa and Air France. The larger A340-500/600 were launched on 8 December 1997; the A340-600 flew for the first time on 23 April 2001 and entered service on 1 August 2002\. Question: Who were the first customers of the Airbus A340?
> Model: Air France and Lufthansa
>
> User: Who is Michelangelo? What are his most famous works?
> Model: Michelangelo was born in 1471 in Florence, Italy. He died in 1526 in Rome, Italy. His most famous works are the Sistine Chapel in Rome and the David in Florence.
> ```
>
> ## 6. "Shapes are Simple"
>
> First, when comparing our results with previous works such as *Text2CAD* and *CAD-Editor*, our generated shapes are not “much simpler,” but rather **comparable**. Second, while some prior works (e.g., *CAD-LLaMA*) include more visually complex shapes, these are mostly unconditional generations, which are inherently simpler because there are no ground truths or constraints on structural fidelity. Third, our paper provides **among the most extensive qualitative samples** in this line of work.
>
> As emphasized, the simplicity primarily arises from the **limited complexity of open-source training datasets**. We anticipate substantial performance improvements once our framework is trained on larger and more sophisticated datasets.
>
> ## 7. LoRA
>
> We tested LoRA ranks of 8, 16, and 32, and observed increasingly better performance with a higher rank. We stopped at 32 because the selected rank already provided adequate performance, and expanding it further would exceed our computational capacity due to the increasing demand on computational costs and GPU memory.
>
> ## Conclusion
>
> Overall, we agree with you that tokenizer-based frameworks hold strong potential to impact the community. We have rigorously tested each component of our system to ensure robustness across scenarios and carefully tuned all hyperparameters. As you kindly noted, our method performs “surprisingly well,” which we believe reflects its non-trivial contribution.
>
> We also respectfully disagree that our work would discourage future research; in fact, we highlight several open directions in the *Limitations* section, including improving reconstruction quality and curating richer datasets. Further adjustments to LoRA configurations or even full fine-tuning are promising future directions.
>
> We would greatly appreciate it if you could **re-evaluate our work** based on these clarifications. Thank you very much for your time and consideration.

---

> ### Comment · Reviewer_uche · 2025-11-19
>
> I want to thank authors for the detailed rebuttal. Here are some additional questions not clear to me and I wish authors could addressed.
>
> 1. Single-token version: This compress a sketch-extrude to one token, which is similar to very early approach like deepcad (which does a more aggressive pooling over all tokens). This baseline is fairly simple and has been demonstrated to not work well, which is clear by its lower scores compared to hnc-cad.
>
> 2.  Comparsion to BPE: As written in my original review,  "A more fair comparision is to convert CAD DSL into text like CADQuery, JSON, or python-like code." The text string in Appendix A.1 is not that much different from DSL besides not assigning special tokens to them. This is not the same as CADQuery or python code, which look something like "s = Sketch().rect(2, 1).vertices().fillet(0.1)". Formulating it in python code will make it more easy for the pretrained model to understand.
>
> 3. The Necessity of Alignment if the LLM Vocab is Expanded: I think this is expected if authors still use rank 32 LoRA. Reserved tokens are not trained as the other tokens, so one needs to optimize the embedding and logit layer together with very high LoRA rank (e.g 256) to make it work. Or a more easy way is to just do full training. Without it it is expected for the model to not learn anything.
>
> 4. Shapes are simple: I disagree that results are due to limited complexity of open-source training datasets. Dataset like WHUCAD is really complicated and demonstrates interesting shapes beyond deepcad.

---

> > ### Author Response · Authors · 2025-11-20
> > **Response to the Additional Questions (Main)**
> >
> > We thank the reviewer for the continued engagement and the opportunity to clarify these specific technical details.
> >
> > ## 1. Single-token version
> > Your understanding is correct regarding the Single-token version. It performs worse than both our tokenizer and hnc, which is within our expectations because it is a much simpler approach compared to the other tokenizer settings.
> >
> > We mention this experiment **in the rebuttal** to demonstrate the baseline performance of a fully compressed, "one-for-all" VQ-VAE. This highlights the contribution of our proposed primitive strategy, which achieves significantly better reconstruction quality (even to hnc) without adding hierarchy to the codes.
> >
> > We mention this experiment **in the ablation studies** as a part of our exploration of **primitive granularity**. This allows us to evaluate how each candidate choice balances reconstruction quality (token fidelity) against the compression ratio (token efficiency).
> >
> > ## 2. Comparison to BPE
> > We understand your suggestion regarding the conversion of CAD data into code-like text formats, such as CADQuery or JSON. Our response is two-fold:
> >
> > First, generating CAD objects using code (e.g., CADQuery, OpenSCAD) \[1-3\] and using procedural steps \[4-9\] represent **two distinct branches** of research. Both are valid approaches supported by established literature, and there is currently **no consensus on which format is absolutely easier for LLMs to learn**. Since our contribution operates within the procedural step paradigm (the latter), we selected baselines and designed our experiments to align specifically with this representation strategy.
> >
> > Second, regarding the specific BPE experiment mentioned in our rebuttal, our goal was to explore whether a standard tokenization strategy (BPE) applied to the *same* CAD sequence would outperform our proposed method. The results of our ablation studies in Table 3 demonstrate that it does not.
> >
> > ## *Ablation studies
> > We **reiterate** that the inclusion of hnc and the specific BPE setup is intended **specifically to support our ablation studies**, which aim to validate our design choices across different training stages. These results serve as **reference points** to help contextualize where our method stands within the broader scope; however, the primary focus of this section remains on the **internal comparison among our proposed variants**.
> >
> > ## 3. The Necessity of Alignment
> > First, we beg to differ regarding the primary cause of the training collapse. Even with a larger LoRA rank or full model fine-tuning, the scarcity of high-quality training data for Text-to-CAD and CAD Editing makes the model prone to overfitting as parameter size increases. We will elaborate on why we consider the data to be the limiting factor in the following section.
> >
> > Second, I believe we both agree that with a lower LoRA rank (actually, 32 is not a small LoRA value) expanding the LLM vocab is expected to be unfeasible. Under this constraint, our method is the **only approach** that remains functional and achieves solid performance. We believe this still represents significant progress for the community, as it offers a viable solution for the majority of domain-specific LLM users who lack the computational resources for full-parameter fine-tuning.
> >
> > ## 4. Shapes are Simple
> > We agree that complex CAD datasets exist in the wild; however, it is important to distinguish between **raw CAD data** and **usable Text-to-CAD or CAD Editing datasets**. The presence of complex geometries does not guarantee the existence of corresponding text-annotated pairs. The bottleneck lies not in the complexity of available shapes, but in the difficulty of converting them into high-quality text-CAD pairs.
> >
> > There are currently two primary methods for creating such pairs: LLM/VLM-based annotation \[5, 6, 8, 10\] and human refinement \[5\]. The latter suffers from prohibitive costs and scalability issues, while the former is restricted by the capability of current VLMs, which often fail to capture the fine details of complex CAD shapes \[5, 6, 10\]. Consequently, existing datasets tend to be noisy and inaccurate regarding complex geometries, which is a fundamental limitation that cannot be resolved simply by adding more generic, unannotated CAD data.

---

> > > ### Author Response · Authors · 2025-11-20
> > > **Response to the Additional Questions (Reference)**
> > >
> > > [1] CAD-CODER: An Open-Source Vision-Language Model for Computer-Aided Design Code Generation. ASME-DETC 2025.
> > >
> > > [2] EvoCAD: Evolutionary CAD Code Generation with Vision Language Models. Arxiv 2510.11631.
> > >
> > > [3] Generating CAD Code with Vision-Language Models for 3D Designs. ICLR 2025.
> > >
> > > [4] Draw Step by Step: Reconstructing CAD Construction Sequences from Point Clouds via Multimodal Diffusion. CVPR2024.
> > >
> > > [5] Text-to-CAD Generation Through Infusing Visual Feedback in Large Language Models. ICML 2025.
> > >
> > > [6] CAD-Editor: A Locate-then-Infill Framework with Automated Training Data Synthesis for Text-Based CAD Editing. ICML 2025.
> > >
> > > [7] FlexCAD: Unified and Versatile Controllable CAD Generation with Fine-tuned Large Language Models. ICLR 2024.
> > >
> > > [8] CAD-Llama: Leveraging Large Language Models for Computer-Aided Design Parametric 3D Model Generation. CVPR 2025.
> > >
> > > [9] CAD Translator: An Effective Drive for Text to 3D Parametric Computer-Aided Design Generative Modeling. ACM MM 2025.
> > >
> > > [10] Text2CAD: Generating Sequential CAD Models from Beginner-to-Expert Level Text Prompts. NeurIPS 2024.

---

> > ### Author Response · Authors · 2025-11-26
> >
> > Dear reviewer,
> >
> > We wanted to briefly reach out to see if you had any further questions or concerns regarding our paper or our previous response. We are fully available and happy to provide any additional clarifications that might be helpful.
> >
> > We appreciate your time and your continuous contribution to the review process.
> >
> >
> >
> > Best regards,
> >
> > Submission 15649 Authors

---

> ### Comment · Area_Chair_r2G1 · 2025-11-27
>
> Reviewer uche,
>
> Could you please check the authors' response to your concerns? The authors have provided new experiments and explanations. An explicit confirmation from you helps me significantly in weighing the decision. Thanks!
>
> AC

---

### Meta-Review · Area_Chair_v2X6 · 2026-01-05

**Summary:**

This paper proposes CAD-Tokenizer, a modality-specific tokenization and decoding framework for unified text-guided CAD generation and CAD editing using a single LLM. The key technical components include primitive-level VQ-VAE tokenization, embedding-space alignment adapters, and FSA-guided decoding to enforce CAD grammar. The paper argues that default text tokenizers are ill-suited for procedural CAD command sequences, and demonstrates through extensive experiments that the proposed tokenizer significantly improves output validity and downstream task performance.

Three of the four reviewers are positive toward the paper and consider it technically sound and potentially impactful. The experimental results consistently support the claimed advantages of the proposed method. One reviewer remains concerned about the fairness of baseline comparisons and the simplicity of the evaluated shapes. Nevertheless, given that the method is technically solid, moderately novel, and supported by empirical evidence, I recommend acceptance.

**Reviewer Concerns:**

Initially, the reviewers raised concerns regarding:
- the fairness of comparisons against existing baselines,
- the positioning and presentation clarity of the paper,
- dataset and shape complexity,
- scalability and potential quantization loss,
- whether task unification is inherently beneficial.

After the rebuttal, most of these concerns were addressed through clarifications, additional experiments, and improved articulation of the paper's scope and claims. The only remaining unresolved objections come from the reviewer who maintained a score of 2.

**Reviewer Scores:**

Three of the four reviewers would raise their scores to positive scores, while the one who gave score 2 keeps the score unchanged. The reviewer also did not engage in the end so it is unlikely he/she will change the score.

---

### Decision · Program_Chairs · 2026-01-26

Accept (Poster)